# Boronation of Biomass-Derived Materials for Hydrogen Storage

Andrea Lazzarini [1] , Alessia Marino [2], Roberta Colaiezzi [1], Oreste De Luca [3,4] , Giuseppe Conte [3,4] ,
Alfonso Policicchio [3,4,5,*] , Alfredo Aloise [1] and Marcello Crucianelli [1,*]

1   Department of Physical and Chemical Sciences, University of L'Aquila, Via Vetoio, 67100 L'Aquila, Italy
2   Department of Environmental Engineering, University of Calabria, 87036 Rende, Italy
3   Department of Physics, University of Calabria, 87036 Rende, Italy
4   Institute of Nanotechnology (Nanotec)—UoS Cosenza, National Research Council, 87036 Rende, Italy
5   CNISM—National Interuniversity Consortium for the Physical Sciences of Matter, Via della Vasca Navale 84, 00146 Roma, Italy
*   Correspondence: alfonso.policicchio@fis.unical.it (A.P.); marcello.crucianelli@univaq.it (M.C.)

**Abstract:** In spite of the widespread range of hydrogen applications as one of the greenest energy vectors, its transportation and storage still remain among the main concerns to be solved in order to definitively kickstart a rapid takeoff of a sustainable $H_2$ economy. The quest for a simple, efficient, and highly reversible release storage technique is a very compelling target. Many studies have been undertaken to increase $H_2$ storage efficiency by exploiting either chemisorption or physisorption processes, or through entrapment on different porous solid materials as sorbent systems. Among these, biomass-derived carbons represent a category of robust, efficient, and low-cost materials. One question that is still open-ended concerns the correlation of $H_2$ uptake with the kind and number of heteroatoms as dopant of the carbonaceous sorbent matrix, such as boron, aiming to increase whenever possible bonding interactions with $H_2$. Furthermore, the preferred choice is a function of the type of hydrogen use, which may involve a short- or long-term storage option. In this article, after a brief overview of the main hydrogen storage methods currently in use, all the currently available techniques for the boronation of activated carbonaceous matrices derived from recycled biomass or agricultural waste are discussed, highlighting the advantages and drawbacks of each of them.

**Keywords:** hydrogen uptake; green economy; activated carbons; sorbent materials; boron doping; spillover

## 1. Hydrogen Storage and Delivery: State-of-the-Art Technology Overview

Increasing global energy demands, limited fossil fuel reserves and production capacities, geopolitical conflicts, and efforts to reduce gas emissions have motivated scientists to put their research efforts into alternative fuels. According to this perspective, among the many green candidates, hydrogen ($H_2$) is becoming an increasingly viable clean option for energy storage and transportation due to its abundance and diverse production sources. For those reasons, governments have started considerably funding relevant research projects [1], and $H_2$ has been the subject of intense focus for the past two decades.

Hydrogen, as a very low emission fuel, can be utilized for many purposes, such as for heating and cooling [1,2], in the transportation sector [3], or for storing excess generated electricity and/or energy, making the possibility of using it subsequently when needed. This scenario should be imagined in the vision of a Hydrogen Economy [4] that would rule the world in the near future. To date, in fact, there is already a huge global market for $H_2$ gas [5] because it is widely used in agriculture for fertilizer production, in refineries for hydrocracking and desulphurization purposes, in food processing, etc. [1], with a portion of the required production, storage, delivery, and utilization infrastructure already in place.

Hydrogen is the most abundant element on earth, but less than 1% exists in the molecular form. The majority is chemically bounded as $H_2O$ in water, and then it is present in liquid or gaseous hydrocarbons.

The clean way to produce it is electrolytically dissociating water by using electricity provided by photovoltaic panels [6] and/or by other novel sources, such as wind or biological processes [7–9], which would present the opportunity for an environmentally friendly energy cycle (Figure 1).

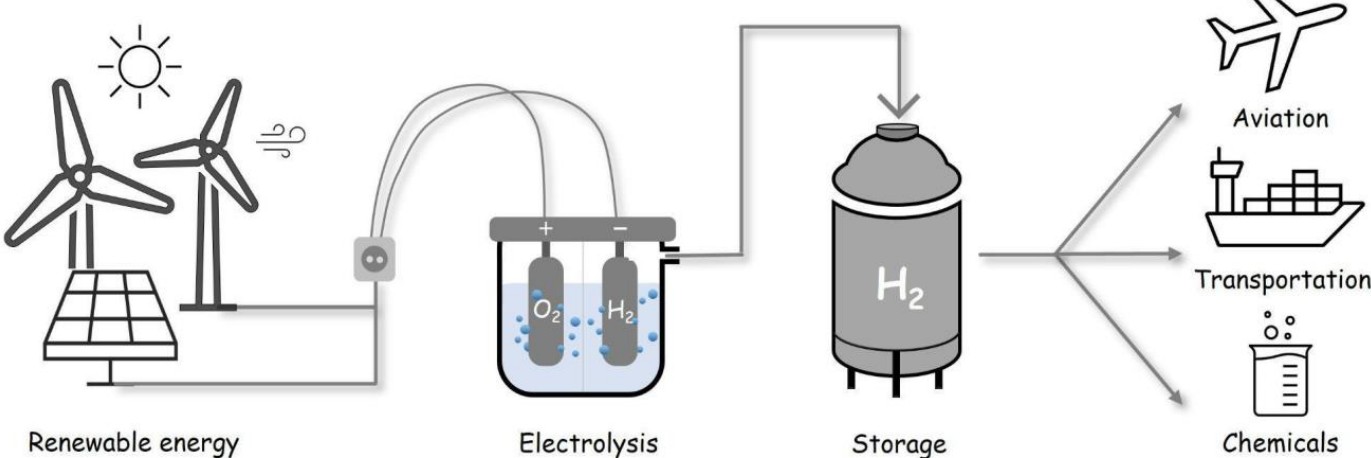

**Figure 1.** Representation of green hydrogen technology.

One feature that makes this element much more energy-efficient than other carriers is its calorific value per mass [6,10], amounting to 39.4 kWh·kg$^{-1}$, which is three times greater than other energy carriers, such as liquid hydrocarbons (13.1 kWh·kg$^{-1}$).

In other words, the energy content of 0.33 kg of $H_2$ corresponds to that contained in 1 kg of petroleum. The difference between the upper and lower heating value is the enthalpy of vaporization of the water that constitutes the product of combustion of this gas when it encounters oxygen in combustion cells. For hydrogen, the upper heating value is reported above, while the lower one is 33.4 kWh·kg$^{-1}$.

In this scenario, hydrogen storage has become a key element in $H_2$ energy systems, especially if we look at its large-scale utilization. Having solid and reliable storage technologies for different applications is a pivotal factor to satisfy the both the current and future demands of the hydrogen energy market [1,2].

Hydrogen storage can be accomplished by various techniques depending also on different applications, such as those of stationary or mobile scenarios [11]. A first method, where there are suitable and available sites, a mid/long-term storage of larger volumes of hydrogen it is possible using geological formations, such as subsurface depleted oil and gas reservoirs, aquifers, or cavern storage [12,13]. Instead, for more immediate application and fast release processes, the most common options are its compression as a gas, its condensation as a liquid, its storage as cryo-compressed gas, or either its chemi- or physisorption by a solid (Figure 2) [14–18]. The essential characteristics of each of the abovementioned hydrogen storage technologies are briefly analyzed.

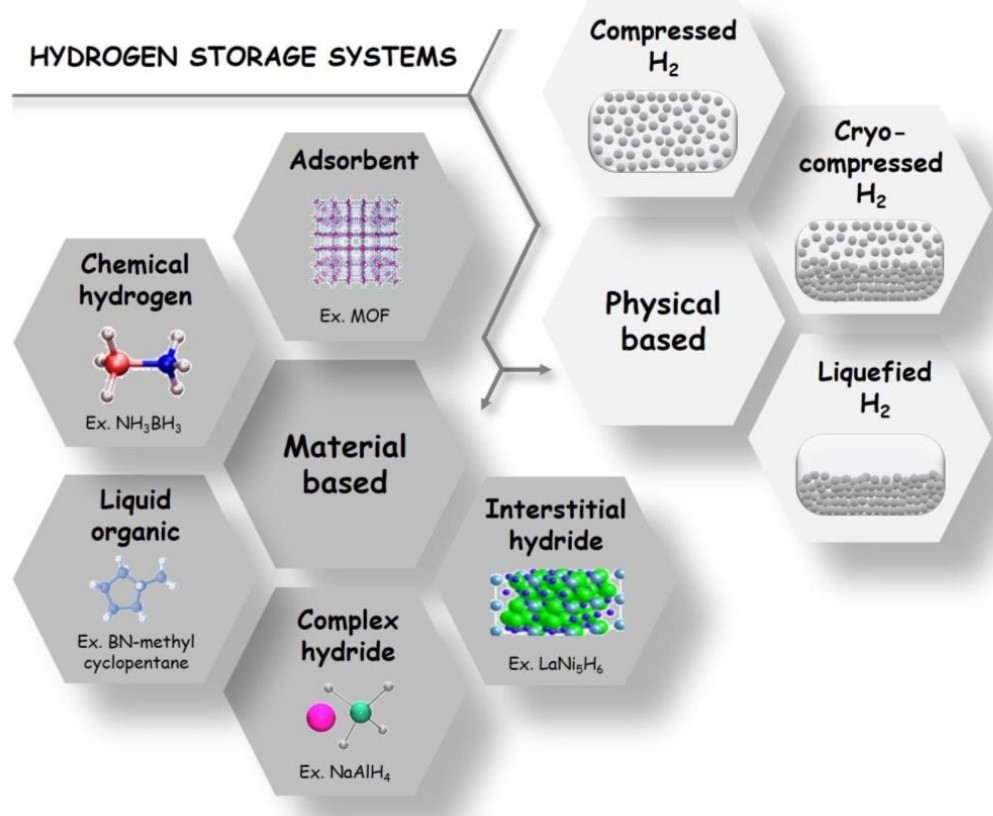

**Figure 2.** Schematization of hydrogen storage technologies.

### 1.1. Compressed Gas

One of the simplest and most common techniques to quantitatively store hydrogen is by compression [10]. The amount that can be stored depends on a balance between the size of the container and the maximum allowable system pressure from a safety perspective. The current state-of-the-art compressed gas container from Quantum Technologies, Inc., is rated for a maximum storage pressure of 70 MPa, allowing hydrogen to reach the volumetric density of 36 kg·m$^{-3}$, which is approximately half compared to liquid conditions and boiling-point temperatures. Therefore, the H$_2$ volumetric density increases by increasing the pressure. On the contrary, its gravimetric density decreases with increasing pressure due to the increase in the thickness of the walls of the cylinder that acts as a reservoir. In fact, safety is a very important issue, and to comply with certain regulatory requirements, these containers must be made of several layers of different materials with a consequent strain on their total mass. The science behind these kinds of vessel, known as TriShieldTM technology, uses aerospace-grade carbon-fiber reinforcement to achieve the safety factor required by current regulations [19]. While this container meets many of the U.S. Department of Energy (DOE) targets [2,20], the high cost of carbon fibers reinforcement is one of the reasons motivating the research for lower-pressure alternatives.

Nowadays, four main types of pressure vessels are being used for storing hydrogen, in a pressure range of 50–100 MPa: (i) fully metallic; (ii) metallic (e.g., steel or aluminum) with high-strength fibers composite overwrap (glass, kevlar or carbon); (iii) full composite wrap with metal liner; and (iv) fully composite pressure vessel, consisting of a polymer as liner and high-strength fibers composites for carrying the structural load [21].

### 1.2. Cryogenic Liquid

The storage of hydrogen as a liquid is a lighter and more compact alternative to the compressed gas. Liquid hydrogen has a high density at low pressure, which reduces the

size of the vessel required for an equivalent amount of fuel [6,10]. Containers for liquid hydrogen storage are frequently aluminum-lined and composite-wrapped with multi-layer vacuum super insulation that reduces heat leak to the liquid [22,23].

As in the compression of hydrogen, the energy balance for its liquefaction must be considered to evaluate its effectiveness as a fuel. In order to produce liquid hydrogen, a substantial energy input is required due to its low normal boiling point (20.4 K). Sensible heat (SH) removal to cool hydrogen from ambient temperature to the boiling point dominates the energy requirement. The enthalpy change associated with sensible heat can be calculated from the enthalpy of gaseous hydrogen at these two temperatures (Equation (1)).

$$\Delta H_{SH} = \Delta H_{g,\, 20.4K} - \Delta H_{g,\, 300K}, \tag{1}$$

The enthalpies at these conditions are $H_{g,\, 20.4K} = 0.29$ MJ/kg and $H_{g,\, 300K} = 4.23$ MJ/kg, which yield $\Delta H_{SH} \cong -3.94$ MJ/kg [24]. The latent heat of vaporization at the normal boiling point of hydrogen is $\Delta H_{VAP} = 0.44$ MJ/kg [25], which is one-tenth of the overall theoretical energy required for liquefaction. The actual process for liquid hydrogen production operates at a Carnot efficiency of <10% and varies with plant throughput [26]. A theoretical minimum energy demand of approximately 40 MJ/kg exists for large-scale plants.

Liquefaction is both a time- and energy-consuming process, and up to 40% of energy content can be lost in the process, as opposed to about 10% energy loss in the compressed hydrogen storage [21]. Thus, this storage method is often used for medium- to large-scale storage and delivery, such as truck delivery and intercontinental hydrogen shipping. Typically, a cryogenic tanker can carry 5000 kg of $H_2$, which is about five times the capacity of compressed hydrogen gas tube trailers [1,2].

### 1.3. Cryo-Compressed Gas

Cryo-compressed hydrogen storage has been introduced to overcome the disadvantages of both traditional storage methods mentioned above (Sections 1.1 and 1.2) by combining their main characteristics [27,28]. It occurs at cryogenic temperatures (20–50 K) on a pressurized vessel, although not as much as with the compressed one ($\cong$35 MPa) [29].

Compared to traditional methods, cryo-compressed $H_2$ storage presents some advantages, such as a higher energy density [30,31], gravimetric capacities and volumetric efficiency [32], a fast filling speed and high-pressure resistance [29,33], a reduced boil-off effect [34], and consequently reduced in-vessel over-pressurization and longer thermal endurance [35]. Additionally it is a versatile method since cryo-compressed storage tanks are designed to endure both very low temperatures and very high pressures [36].

Nevertheless, there are still some limitations preventing this technology from becoming commercially viable. In particular, cryo-systems are usually complex and hard to implement, requiring the careful and permanent management and monitoring of their thermal insulation levels [32], as they need high energy for operation, have considerable maintenance costs, and have a short no-loss unused period [37].

### 1.4. Material-Based $H_2$ Storage

Hydrogen can be retained from solid state sample either as dissociated atoms, which therefore can be chemically attached by covalent bond (chemisorption), or as whole molecules, and in this way be physically withheld by van der Waals–like forces (physisorption) [38].

In detail, in the case of chemisorption, the interaction between $H_2$ and the solid sample can occur either in the bulk or on the surface. In particular, intermetallic alloys enable the catalysis of the $H_2$ molecule, resulting in the breaking of the bond between the hydrogen atoms with a consequent penetration of the resulting H atoms into the crystal structure of the material, thus affording the formation of metal and/or complex hydrides [1,39]. On the contrary, in the physisorption, the interaction between $H_2$ and the solid material occurs without any molecule modification, with adhesion on an adsorbent surface being easily removable (enrichment of one or more components on an interface).

The difference between the processes described above lies in the energies involved. In order to quantify the latter, the enthalpy of formation ($\Delta H$) is considered as a reference parameter. In the case of chemisorption, $\Delta H$ is particularly high and lies in the range of (10–100) kJ/mol, while the physisorption involves enthalpy in the range of (1–10) kJ/mol.

Physical bonding can therefore be created at cryogenic liquid nitrogen temperatures and high pressures (0–100 bar), while the chemical bonding can occur at ambient temperatures and pressures.

An important issue is related to the release of the stored hydrogen: usually, the chemisorption needs energy by thermal annealing ranging between room temperature [10] and 900 K [40] in order to remove the chemisorbed atoms. In several cases, the chemisorption mechanism results even in an irreversible process. On the contrary, physisorption is always a reversible mechanism [14,18] in which an increase of temperature of about 10 K and/or depressurization are enough to get back all the $H_2$ previously stored. Considering the different bond energies and the surface barrier, the kinetics of the hydrogen molecules go from fast to slow diffusion in the physisorption and chemisorption, respectively.

What penalizes chemical storage is therefore the energy expenditure in the desorption phase; the emergence of hysteresis, which indicates an irreversible change in the material structure; the low gravimetric density due to the high density of the absorbing materials; the weight of these materials; and their thermal instability. The percentages by weight with this storage technology range from 1 to 12.7% (lithium hydride) compared with 1% for common cylinders. At present, the materials available for chemical storage are too heavy: e.g., at equal weight, a vehicle has three times less range than cylinders with compressed or liquid hydrogen. In contrast, there are undoubted advantages in terms of convenience, compactness, storage stability, and safety.

The $H_2$ storage by physisorption also has lights and shadows: in fact, the necessity to recur to cryogenic temperatures and high pressures is certainly a disadvantage in terms of the energy balance of the whole operation. On the other hand, there is no need for additional heating for the rapid $H_2$ desorption, and, moreover, the thermal and structural stability of the materials for successive cycles of adsorption and desorption is combined with their good environmental impact.

Generally, in daily use, management of the heat produced/absorbed during hydrogen charging and discharging processes becomes a critical issue, and the characteristics of the used adsorbent materials greatly contribute to this.

In both chemisorption and physisorption processes, some of the adsorbent materials are initially powdered and/or liquid, such as liquid organic hydrogen carriers (LOHCs) [41,42]. Nevertheless, as such, they are not efficient phases as far as heat transfer is concerned, and for this reason, base materials are often preprocessed by melting, coating, templating, and uniaxial pressing methods and then placed in the reservoir [39]. In addition, a heat exchanger for thermal management and connections for $H_2$ flow control and filtration at the inlet and outlet of the reservoir are also often placed inside the latter [43].

### 1.4.1. Intermetallic Compounds as $H_2$ Storage Systems

Intermetallic compounds are well-known for their hydrogen uptake capacities [44–47]. They are often used as calibration or validation materials for volumetric or gravimetric sorption techniques. In practical application, the compounds would likely be deployed in powder or pellet form in a pressure vessel to lower the storage pressure for a given quantity of $H_2$ [48–50]. Table 1 lists some of the common metal composites, along with their absorption capacities, as well as some recently discovered high-capacity materials. Note that the reported capacity is a function of pressure and temperature, which can vary widely for each material.

**Table 1.** Storage capacity [51–53] and ΔH values [52,54,55] of several intermetallic compounds.

| Metal(s) | Hydride | Capacity (wt%) | $P_{eq}$ (MPa) | T (K) | ΔH (kJ/mol) | Refs. |
|---|---|---|---|---|---|---|
| Pd | $PdH_2$ | 0.56 | 0.002 | 298 | 41 | [51,52] |
| Mg | $MgH_2$ | 7.6 | 0.19 | 573 | 75.4 | [51,52,54] |
| Ti | $TiH_2$ | 4.0 | 0.09 | 909 | 125.6 | [51,52] |
| V | $VH_2$ | 2.1 | 0.37 | 313 | 58.6 | [51,52] |
| FeTi | $FeTiH_{1.94}$ | 1.89 | 0.5 | 303 | 28 | [51,52,54] |
| $LaNi_5$ | $LaNi_5H_6$ | 1.37 | 0.2 | 298 | 31 | [51,52,54] |
| $Mg_2Ni$ | $Mg_2NiH_4$ | 3.59 | 0.1 | 555 | 65 | [51,52,54] |
| NaAl | $NaAlH_4$ | 8.0 | 9 | 403 | 113 | [51,52,54,55] |
| $Ti_{0.9}Al_{0.06}V_{0.04}$ | $Ti_{0.9}Al_{0.06}V_{0.04}H_2$ | 3.8 | 0.1 | 300 | NR | [51–53] |

The pressure–concentration behavior can influence the effectiveness of a material in a proposed storage system and often forms the basis for the selection. Despite their relatively high storage capacities, metal hydrides have not seen widespread practical use because they often require a steep energy input to extract the absorbed hydrogen. Formation energies are often in the range of 28–126 kJ/mol, as shown in Table 1. An additional consideration for practical application is the kinetic response of the materials, which is often quite slow. The equilibration time needed is often approximately several hours for both the uptake and release of the stored hydrogen.

Some intermetallic compounds are also rather sensitive to gas-phase impurities, demonstrating reduced capacity, slower kinetics, or both [56]. Other intermetallic compounds, as composites or systems doped with other materials, are quite promising, demonstrating improved kinetics and requiring less rigorous conditions to recover the stored hydrogen [57–59]. Transition metals that form hydrides are one of the several sources of hydrogen atoms which can spillover [60], that is, migrate from the metal particles to the surface support and, eventually, into the bulk material (Figure 3) [61–63].

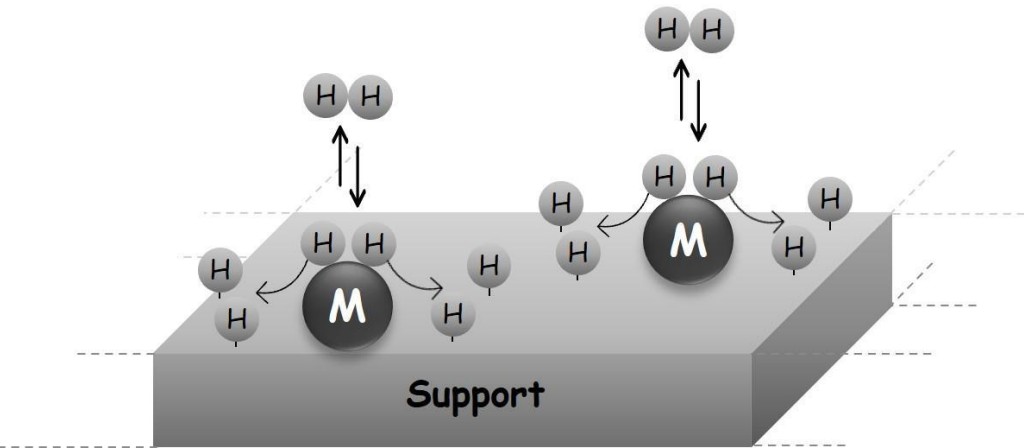

**Figure 3.** Schematic representation of spillover mechanism.

1.4.2. Porous Solids as $H_2$ Storage Systems

Porous-material-based storage systems are potentially a means to achieve reliable and high-capacity storage units [1,2]. Many porous solids have been investigated as sorbent systems for hydrogen storage in the last decades [64,65], including zeolites [66], porous silica [67,68], porous clay [69,70], metal–organic frameworks (MOFs) [71,72], nanoporous polymers [73], carbon nanostructures [74,75], and activated carbons [76,77], each of which presents different merits and shortcomings, which are summarized in Figure 4 [78].

Among the different drawbacks that many available materials up to now show in relation to different type of applications, we can refer to slow desorption/sorption kinetics,

unstable structures, relatively high thermal stability, high weight, irreversibility on cycling, and expensive production costs.

For these reasons, research has been focused on the development and the analysis of sorbent systems (meso- and nanostructured) which are frequently characterized by two important and well-known parameters [79], namely high values of Brunauer–Emmett–Teller Specific Surface Area ($S_{BET}$) and pore volume (PV) [80]. The pore volume is further divided into micropore (<2 nm) and mesopore volumes ($2 \leq d \leq 50$ nm) since the pore dimension is critical to determining the strength of interaction. Large micropore volumes contribute to high $S_{BET}$ and increased interaction energy between the adsorbent and adsorbate. The optimal pore size distribution (PSD) allows for maximum interaction (capacity) and easy transport (kinetics) based on the dimensions of the adsorbate, and so it gives the basis of the selection of an adsorbent for targeted applications. This information, properly tuned, is a key parameter for the optimization of new materials able to enhance the maximum interaction (capacity) and improve kinetic response.

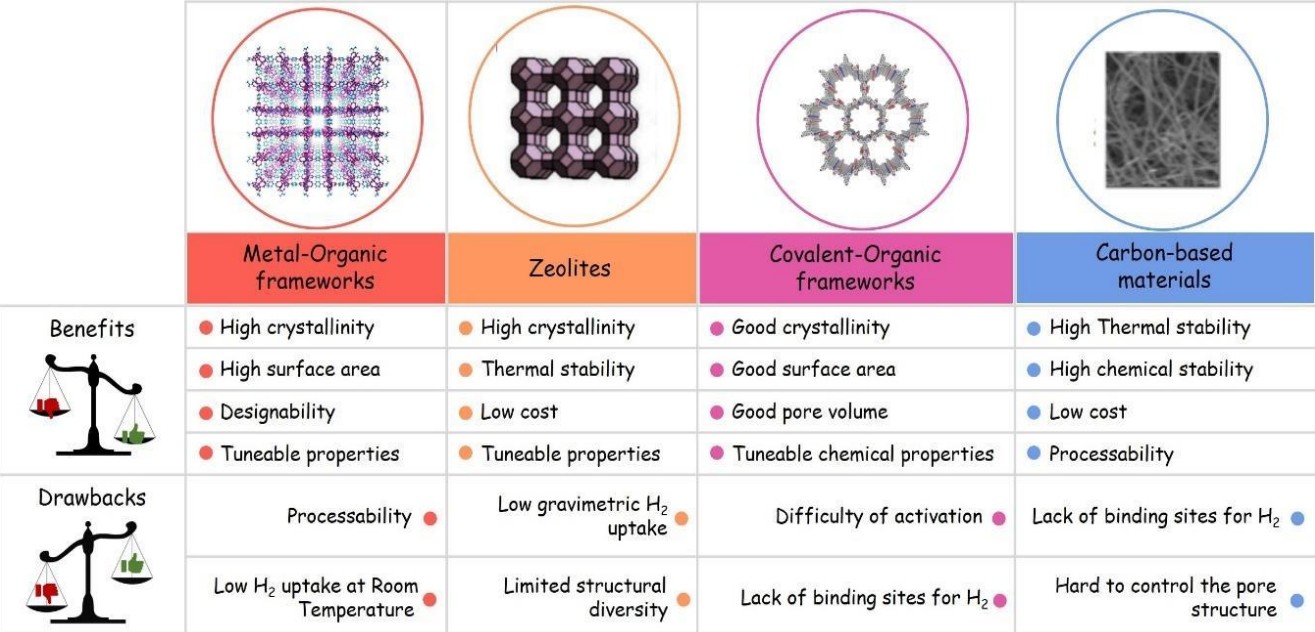

**Figure 4.** Summary of benefits and drawbacks of the main sorbent systems for hydrogen storage.

Inter alia, activated carbons (ACs) are particularly suited for large-scale applications because of the abundance and low cost of the raw materials from which they are made [81–83], the ease of synthesis process that allows researchers to tune the pore structure, their sustainable regeneration [84], and their chemical stability [85].

## 2. Biomass-Derived Carbonaceous Material

Carbon has an atomic number of 6, and its ground-state configuration is [He] $2s^2$ $2p^2$. Like other p-block elements, solid elemental carbon exhibits the phenomenon of allotropy. One of the s electrons can be promoted and hybridized with different numbers of p-orbitals to give rise to three types of hybrid orbitals: $sp^3$, $sp^2$, and sp. These are at the origin of the three basic carbon structures, respectively diamond, graphite, and carbynes [86]. Some of the allotropes, e.g., diamonds, have a rather limited relevance to adsorption and are not in the aim of this work, so from now on, we focus our attention only on the carbon structures with relevant adsorption characters. In the last twenty years, many experimental works have been carried out to achieve a complete picture of what enhances or reduces the adsorption capacity of carbon nanostructured materials in order to reach the DOE targets [2,87].

Porous systems (e.g., nanotubes, nanofibers, activated carbons, activated fibers, carbons from templates, powders, doped carbons, etc.), compared to liquid and gaseous media, offer the advantage of lower-pressure hydrogen storage, design flexibility, increased safety, and reasonable volumetric storage efficiency. However, to date, the technology is not yet mature, and there are still no forthcoming solutions for avoiding weight/cost penalties and tackling thermal management issues associated with this option [88].

Materials with a large specific area, such as activated carbon, nanostructured carbon, nanofibers, and carbon nanotubes, are promising substrates for physisorption. In terms of storage capacity, curved structures appear to be more efficient as compared to high and flat surface area (e.g., graphite), showing up to 25% extra capacity at low temperatures [6]; thus, the capacity decreases as the temperature increases. This behavior is due to the greater attractive forces compared to the open flat-surface structures [11].

In recent years, growing research interest has been focused on activated carbons' production from biomass (Table 2), starting from various agricultural wastes, due to their high adsorption capacity, considerable mechanical strength, and low ash content [89]. Some of the agricultural wastes include the shells and stones of fruits and wastes resulting from the production of cereals, bagasse, and coir pith [90].

**Table 2.** Biomass-derived ACs, activation methods, and related hydrogen storage capacity.

| Precursor | BET Surface Area (m²/g) | Activation Method | P (bar) | T (K) | H₂ Storage (wt%) | Refs. |
|---|---|---|---|---|---|---|
| Coffee-bean wastes | 2070 | KOH | 120 / 40 | 77 / 298 | 4.0 / 0.6 | [91] |
| Peanut shell | 1726 | KOH | 100 | 298 | 1.2 | [92] |
| Coconut shell | 2800 | KOH | 100 | 297 | 0.8 | [93] |
| Corncob | 3012 | KOH | 1 / 50 | 77 / 298 | 2.0 / 0.4 | [94] |
| Rice hull | 3969 | NaOH/$\Delta$T | 12 | 77 | 7.7 | [95] |
| Biomass wood | 2450 | $H_3PO_4$/KOH | 20 | 298 | 0.8 | [96] |
| Bamboo | 3148 | KOH | 1 / 40 | 77 | 2.7 / 6.5 | [97] |
| Sword-bean shells | 2838 | KOH/$\Delta$T | 1 / 40 | 77 | 2.6 / 5.7 | [98] |
| Beer lees | 1927 | KOH | 1 | 77 | 2.9 | [99] |
| Melaleuca bark | 3170 | KOH | 10 | 77 | 4.1 | [100] |
| Coconut shells | 415 | $O_2$/573 K/6 h | 40 / 100 | 298 | 0.3 / 0.5 | [101,102] |
| Lignin | 1000 | $CO_2$/1273 K/1 h | 1 | 77 | 1.8 | [103] |
| Cornstalks | 3200 | KOH | 40 | 77 | 4.4 | [104] |
| Olive stones | 1269 | KOH | 200 | 77 | 6.0 | [105] |
| Palmyra sprouts | 2090 | KOH | 15 | 298 | 1.1 | [106] |
| *Cannabissativa* L. | 3241 | KOH | 1 | 77 | 3.3 | [107] |
| Fungi-based chars | 2500 | KOH | 1 / 35 | 77 | 2.4 / 4.7 | [108] |
| Chitosan | 3066 | KOH | 1 / 40 | 77 | 2.9 / 5.6 | [109] |
| *Neolamarckia cadamba* | 3462 | KOH | 1 | 77 | 2.8 | [110] |
| *Posidonia oceanica* | 2800 | KOH | 80 | 77 | 6.3 | [111] |
| Pinecones | 1173 | KOH | 1 / 80 | 77 | 1.6 / 5.5 | [81] |
| Empty fruit bunch | 687 | KOH/$CO_2$ | 19 | 77 | 2.1 | [112] |
| Rice husk | 1490 | $H_3PO_4$/$\Delta$T | 10 | 77 | 1.8 | [113] |
| Lychee trunk | 3400 | KOH / KOH–Pd (10 wt%) | 1 / 60 | 77 / 303 | 2.9 / 0.5 | [114] |
| Celluloseacetate | 3800 | HTC | 20 / 30 | 77 | 8.1 / 8.9 | [115] |

It has been observed that, among the various agriculture residues, tamarind seeds are found to be interesting, as they are available in various forms, viz seeds, seed husk, kernel powder, etc. Tamarind kernel powder is normally used in industries such as textile, dying and printing, jute, cardboard, mosquito coil, food additive, oil-well drilling, paper industry, etc. [116,117]. The botanical name is *Tamarindus indica* L., and India is the major producer of tamarind on a large scale. However, there are still few published works on the preparation of high-surface ACs by using hemp (*Cannabis sativa* L.) stem as raw materials.

As an extension of the agricultural-wastes-based ACs, it is thought to use hemp stems to synthesize activated carbon materials with super-high surface areas.

Hemp (*Cannabis sativa* L.), an annual herbaceous plant, has been planned agriculturally for many centuries for its bast fibers and hempseed oil [118]. Due to the increasing demand for hemp fibers for clothing and the development of technologies for hemp fibers' processing, the cultivated area of hemp has increased remarkably. China currently cultivates hemp over an area of around 20 thousand hectares, and the planting area increased up to about 670 thousand hectares in 2020 [107]. In the processing of hemp fibers, the hempseed, hemp bast, and hemp stem become interesting by-products. The hemp seed and hemp bast can be used in the production of biodiesel [119] and activated carbon fibers [120], respectively. However, there are still few works on the preparation of high-surface ACs by using hemp (*Cannabis sativa* L.) stems as raw materials. As an extension of the agricultural-wastes-based ACs, it is thought to use hemp stems to synthesize activated carbon materials with super-high surface areas.

Rosas et al. [121] prepared a hemp-derived AC monolith with a surface area of 1500 $m^2/g$ via chemical activation. However, the AC monolith's physical properties (in terms of significant mesoporosity and relative low surface area) have not proved to be sufficient for improving gas adsorption capacity. In addition, there are few works focusing on the preparation and porous structure characterization of hemp-based high-surface-area ACs. As a consequence, the utilization of hemp stem to produce ACs not only makes a considerable economic benefit to meet the increasing demand of ACs in different application areas, such as food, environment, chemistry, and energy [122], but also makes a contribution to the theoretical research on the properties of carbonaceous materials derived from agricultural wastes.

## 2.1. Activated Carbons (ACs)

The most common carbon materials for industrial use have been classified as active (or activated) carbons (ACs), owing to the procedures used to "activate" or modify the structure of the adsorbent for a specific application. Indeed, it is the facility to tailor the adsorbent properties that supports its broad range of applications [123–127]. The process of ACs' production begins with the selection of a raw carbon source which is selected based on design specifications since different raw sources will produce ACs with different properties, e.g., a pore structure that can vary both in shape and in dimensions, with pores ranging from less than 1 nm to more than 1000 nm (Figure 5). The result could be a highly nanostructured porous carbon containing crystallized graphite and/or amorphous carbon with a high specific surface area. ACs have been considered a potential candidate for hydrogen storage because of their very good structural and chemical properties, together with their ability to adsorb gas in a fully reversible way and connected also with their characteristic to be relatively cheap and accessible on a commercial scale.

The production of activated carbon, traditionally, has been obtained from coal, lignite, wood, petroleum residues, and polymers, all of which are very expensive and, except for wood, non-renewable. Consequently, in the search for renewable and cost-competitive alternatives, in the recent years, research has grown focusing on the production of activated carbon from by-products and biomass residues (so-called activated biochar) [128–130].

In the literature, since 2000, many precursors have been mentioned for ACs' production, such as corn, rapeseed, barley, bagasse, bamboo, scrap tires and saw dust, almond, pecan, English walnut, black walnut and macadamia nut, pistachio, hazelnut shells, rice husk, rice bran, coconut shells, etc. [125,131]. Their preparation most often includes dehydration, carbonization, and activation.

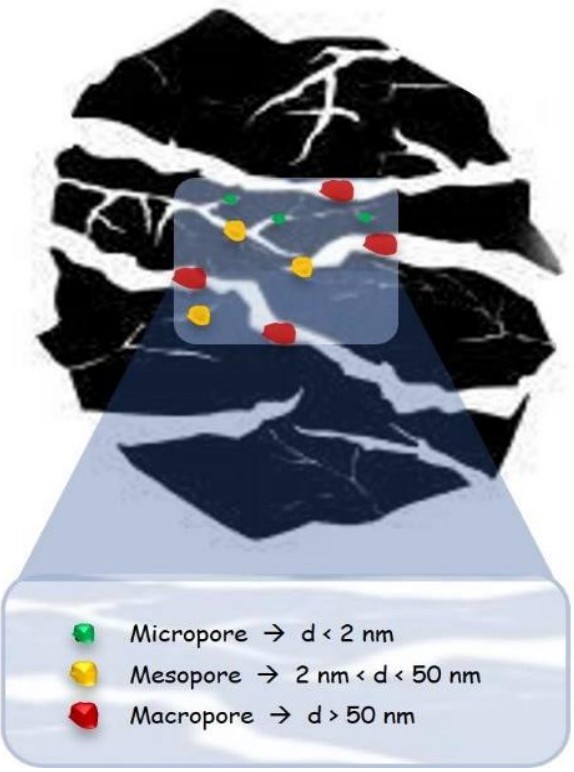

**Figure 5.** Illustration of different types of pores in ACs particles.

Dehydration and carbonization involve slow heating of the source in anaerobic conditions. Chemicals such as zinc chloride or phosphoric acid can be used to enhance these processes. The stage of activation requires exposure to additional chemicals or other oxidizing agents such as a mixture of gases.

In general, raw materials are carbonized at temperatures up to ~1000 K to remove volatiles [132]. Then the product is activated by gaseous or chemical techniques to develop porosity and to increase the surface area. Gas-phase methods are performed in oxygen, carbon dioxide, or steam at temperatures up to 1173 K [132]. The type of gas phase and temperature impact the pore structure of the final product. Oxygen is an aggressive activation species, gasifying carbon at a rate five orders of magnitude greater relative to carbon dioxide or steam, with the latter molecules frequently being used for the mild activation and improvement of a uniform pore structure.

Chemical methods often involve the addition of inorganic species in solution, which subsequently act on the precursor materials prior to or during carbonization. An activation technique commonly known is based on the use of alkali hydroxide (KOH, NaOH), either in solid form or dissolved in aqueous solutions.

A complete study of various kinds of ACs has been performed in the last ten years. Below, we report a series of research works subdivided by preparation methods, where, in addition, the authors also tested the performances in terms of hydrogen storage.

### 2.1.1. ACs by Physical Activation

Generally, published studies concerning the production of activated carbons (ACs) through physical activation techniques are focused on the use of residual biomass sources, because it has been observed that their use as precursor materials can either match or improve the properties of commercial activated carbon obtained from traditional sources [125], in particular, optimizing the activation process by using carbon dioxide or steam [89,133–136]. Furthermore, although in smaller numbers, other studies have analyzed the influence of the carbonization process conditions in the subsequent activation stage [137–141].

The aim is, hence, to tailor pore structures and textures of the porous carbon materials to obtain the high surface area and micropore volume that are essential for enhancing the hydrogen storage capacity [142–145]. A comprehensive survey of the literature showed that an "ideal" biomass-based AC sorbent with a specific surface area of 4000 m$^2$/g or more, a micropore volume greater than 1.5 cm$^3$/g, and a micropore size typically below 1 nm (smaller pores benefit by stronger physisorption due to the high heat of adsorption) may help in increasing H$_2$ uptake, reaching a value higher than 10 wt%, at cryogenic conditions [146].

Studies on the use of activated carbons for hydrogen storage started to appear about thirty years ago [147], and a large variety of activated carbons, on a wide range of pressures and temperatures, have been investigated [101,115,148].

In 1980, Carpetis and Peschka [149] reported on the H$_2$ storage capacity of several adsorbents, including activated carbons, in different temperature and pressure ranges, respectively, from 65 to 150 K and from 0 to 41.5 bar, and they observed a maximum nominal uptake of 5.2 wt% at 65 K and 41.5 bar. This work gave an early indication of the potential of ACs as H$_2$ storage, although there was still no clarity on the key parameter involved in the adsorption process.

Recently, Sultana et al. [146], studying factors affecting H$_2$ storage in biomass-derived AC, highlighted, once again, the significance of a specific surface area for a substantial H$_2$ uptake, showing, for example, how the storage capacity can increase from 2.1 to 8.9 wt% with the increase of surface from 687 to 3771 m$^2$/g. In fact, to cite only a few examples, AC obtained by using hemp stem [150] and melaleuca bark [100], as starting materials, exhibit a trend of an increasing quantity of H$_2$ adsorbed in the range of 1.57–3.28 wt% and 1.21–3.92 wt% for a corresponding surface of 922–3241 m$^2$/g and 1092–2986 m$^2$/g, respectively. However, as previously mentioned, specific surface area is not the only key parameter for excellent performance; in fact, the latter must be coupled with adequate porosity/volume ratio of pores and/or surface functionality. Materials with a similar surface area can, in fact, show different results, as observed by Ding et al. [151] and Ujjal Kumar Sur [152].

More evidence arises from the work of Balathanigaimani et al. [99] wherein the H$_2$ uptake capacity of beer-lees-derived ACs decreased (from 2.92 to 2.43 wt%) with the increase in surface area (from 1927 to 2408 m$^2$/g), this trend being otherwise in accordance with increasing micropore volume and microporosity of the support. Chen et al. [98] reported on H$_2$ adsorption on sword-bean-shell-derived ACs, showing an uptake capacity of 2.25 wt% despite having a very similar surface area of the beer-lees-derived ACs reported from Balathanigaimani et al. (specific surface area (SSA) of 1927 vs. 1930 m$^2$/g, respectively) [99]. In the hemp-stem-derived ACs of Zhang et al. [150], the H$_2$ uptake barely enhances (from 1.57 to 3.28 wt%) despite a tripling of the surface area (from 922 to 3241 m$^2$/g). Coconut-shell-activated carbon analyzed by Nasruddin et al. [101], at ambient condition (298K) and high pressure (40 bar), showed an H$_2$ uptake of around 0.25 wt% connected to an SSA of around 415 m$^2$/g in the same conditions but moving to higher pressure (100 bar) [102]. Activated carbon samples obtained by using the same precursor (coconut shell) but under different activation conditions (steam at 1173 K) were analyzed by Zhao et al. [153]. They investigated a temperature range from 77 to 114 K and pressure up to 1 bar, obtaining H$_2$ adsorption value close to 2.2 wt%.

Briefly, as shown above, various physical (e.g., SSA and pore size and volume) and chemical factors (possible presence of heteroatoms such as N and O, especially for biocarbons coming from agricultural wastes) can have more or less influence in determining the actual hydrogen uptake capacity for ACs.

Among the most recent works concerning the production of ACs by physical activation, Rowlandson et al. showed materials produced from lignin having an SSA > 1000 m$^2$/g and a hydrogen storage capacity of 1.8 wt% at 77 K, under 1 bar [103]. The authors highlighted that there are no substantial differences in the textural and hydrogen adsorption properties obtained by varying the starting raw material. They also showed a pyrolysis process with

relatively low temperatures for industrial processes, thus making large-scale production more efficient.

In the literature, many studies combine physical activation ($CO_2$ and/or steam) and chemical activation ($ZnCl_2$, KOH, NaOH, etc.) of different kind of biomass in order to improve textural properties for hydrogen storage and/or energy-related applications [147,154–157].

### 2.1.2. ACs by Chemical Activation (KOH)

This activation process develops an extensive microporous structure in the carbon materials that contributes to high surface areas (3000 $m^2$/g or more), leading to the classification of "superactivated" carbons.

Usually, the chemical activation process may follow a first carbonization step. This procedure was used by Zhang et al., who synthesized an activated carbon with a high surface area and large micropore volume derived from cornstalks and after KOH activation [104]. The microporous carbon showed a favorable $H_2$ storage performance with a maximum $H_2$ adsorption capacity of 4.4 wt%, at 77 K and up to 40 bar. Such a high adsorption capacity and good selectivity for $H_2$ could be attributed to its abundant micropore structure and high surface area. Similar adsorption capacities, even if at lower pressures, were recorded by Xiao et al., who fabricated porous carbons with tunable morphologies and texture, from melaleuca bark waster biomass, activated by KOH [100]. The resulting samples demonstrate both a high surface area (up to 3170 $m^2$/g) and large hydrogen storage capacity (4.08 wt% at 77 K, 10 bar), implying their great potential use as hydrogen storage materials.

Bader and Ouederni [105] studied the performance of five biomass-based carbon samples derived from olive-stone wastes on hydrogen storage. All samples were prepared through a KOH activation procedure. Differences in the microstructures between the prepared materials were obtained by varying the KOH/precursor weight ratio from 1:1 to 5:1, with the aim of shedding light on both the optimization of their hydrogen storage capacity and the clarification of the adsorption mechanism. The increase of the activation ratio leads to the changing of the nature of the carbon from microporous to micro-mesoporous. Consequently, the hydrogen adsorption capacity was enhanced at high $H_2$ pressures due to the increased surface area. Hydrogen storage capacities of 6.0 wt% (77 K, 200 bar) and 1.22 wt% (298 K, 200 bar) were reached at the KOH/precursor weight ratio of 4:1, making this biomass-based carbon sample promising for $H_2$ storage applications.

Using the same biomass as a precursor, Schaefer et al. [158] reported the effect of heteroatoms on the hydrogen adsorption properties of activated and hybrid carbon materials. Olive stones were activated chemically with KOH, subsequently washed or not, and then eventually oxidized with ozone. In addition, a series of activated carbons prepared by the chemical activation of sucrose was also studied. As a result, many activated carbon materials having different structural properties, i.e., pore-size distributions, surface areas, average micropore widths, oxygen contents, and amounts of mineral matter, were compared, and many correlations between textural parameters, composition, and adsorption properties were evidenced and critically discussed. This study clearly pointed out that the hydrogen uptake at 77 K is controlled by the following parameters, having a decreasing role: SSA, average micropore size, surface chemistry, and shape of the pore-size distribution (PSD). The adsorbed hydrogen uptake was in the range of 0.19–0.42 wt% at 298 K and 100 bar. In addition, the presence of large amounts of alkali metals can further improve the hydrogen adsorption properties, even if the surface chemistry still has the most relevant role, especially through the acidic surface functions. Indeed, as stated by authors, the strong electron-acceptor character of acidic functions—in particular, carboxylic acids—along with the difference of electronegativity between carbon and oxygen which may affect the formation of partial charges and polarized physisorption [159], should play a major role in hydrogen adsorption.

Another raw precursor to synthesize heteroatom self-doped activated bio-carbons is the sword-bean shell, as reported by the already-cited T. Chen et al. [98], using KOH activation. The ratio between micropores and total pore volume in the prepared samples

ranges from 0.55 to 0.75, indicating a development of highly microporous activated carbon. The increase of the weight ratio between KOH and bio-carbons leads to an improvement of the structural properties of the carbon material, which showed an $S_{BET}$ value up to 2838 m$^2$/g and a cumulative pore volume of 1.54 cm$^3$/g. These structural improvements enhanced absorption capacities at 77 K up to a maximum of 2.63 wt% at 1 bar and 5.74 wt% at 40 bar, respectively.

Even more interesting results were reported in the study of Samantaray et al. [106] wherein an activated carbon material was derived by using palmyra sprouts as a biomass precursor, with KOH used for the activation process. High surface area and pore volume values of 2090 m$^2$/g and 1.44 cm$^3$/g, respectively, were found. The hydrogen uptake capacity of about 1.06 wt% at 298 K and 15 bar makes the palmyra-sprout biomass promising for its use as a catalyst support material for improving the hydrogen storage by means of a spillover mechanism [60].

Yang et al. [107] studied a series of chemically (KOH) activated carbons (ACs) derived from hemp (*Cannabis sativa* L.). The maximum hydrogen storage capacity is 3.28 wt% at 77 K and 1.0 bar for an AC material with a surface area of 3241 m$^2$/g and total pore volume of 1.98 cm$^3$/g. The microporous structure of the activated carbon material plays the main role for the hydrogen adsorption; however, the mesopores, ranging from 2 to 5 nm, also make an important contribution. Indeed, the gas adsorption amount is dominated by ultramicropores at lower pressure, while larger micropores and mesopores make major contributions at a higher pressure.

Wang et al. [108] reported the synthesis of a set of porous carbons prepared by chemical activation of various fungi-based chars with KOH. The activation parameters, i.e., temperature and KOH amount, played a leading role in regard to the porosities of the resulting porous carbons (mainly consisting of micropores), whereas the kind of fungi used as a precursor had a marginal impact. The surface areas and pore volumes values obtained after the chemical activation ranges from 1600 to 2500 m$^2$/g and from 0.80 to 1.56 cm$^3$/g, respectively. A uniform micropore structure, with a size of 0.8–0.9 nm, is common for all the resulting activated carbons, while some porous carbon materials have another set of micropores ranging from 1.3 to 1.4 nm, which can be further broadened to 1.9–2.1 nm by the increasing of either the activation temperature up to 1023 K or KOH/char mass ratio up to 5/1. An excellent hydrogen uptake of up to 2.4 wt% at 77 K and 1 bar was achieved for these fungi-based porous carbons, while at 77 K and 35 bar, the adsorption capacity ranged from 4.2 to 4.7 wt%.

Highly microporous ACs were successfully prepared using chitosan as a carbon precursor activated chemically with KOH in different experimental conditions [109]. The carbonization and activation temperatures, as well as the KOH/char ratios, strongly affected the porous structure developed during activation. Surface areas and pore volumes were in the range of 922–3066 m$^2$/g and 0.40–1.38 cm$^3$/g, respectively, while microporosity was the main contributor to the total pore volume for all ACs. As a result, the activated carbon materials exhibited excellent hydrogen uptakes at 77 K, i.e., 2.95 wt% at 1 bar and 5.61 wt% at 40 bar. Micropores with a size ranging from 0.7 to 1 nm played a leading role for hydrogen adsorption at atmospheric pressure, while larger pores exerted a marginal influence. Conversely, adsorption at 40 bar was enhanced by the presence of supermicropores and small mesopores. All of these results indicate that chitosan-based ACs can be used in hydrogen storage applications.

Hu et al. [110] synthesized activated carbon materials from Neolamarckia cadamba, followed by KOH chemical activation. The porous carbon materials exhibited high values of $S_{BET}$, ranging from 2743 to 3462 m$^2$/g, and large pore volumes that varied from 1.09 to 1.67 cm$^3$/g, as well as a PSD that ranged from 0.5 to 3.7 nm. The maximum hydrogen adsorption capacity of 2.81 wt% at 1 bar and 77 K was reached for the activated porous carbon prepared at 1073 K and with a KOH/carbonized material ratio of 3.

Pedicini et al. [111] pyrolyzed at 873 K Posidonia Oceanica and wood chips as raw storage-system precursors. After this step, the biochar obtained from the carbonization

was chemically activated with KOH. The activated carbons exhibited a specific surface area of 2800 m$^2$/g, and the corresponding adsorption values were about 6.3 wt% at 77 K and 80 bar, much higher than the values provided by the Chahine rule, which establishes the relationship between H$_2$ storage capacity and the specific surface [160–163]. In this specific case, the Chahine rule estimates for these absorbing surfaces a value of 5.6 wt%. It is worth highlighting that 80% of the final tested absorption value is already reached at 10 bar.

Stelitano et al. [81] used common pinecones as a biomass precursor. The chemical activation given by KOH induced the formation of an optimal pore-size distribution for hydrogen adsorption, centered at about 0.5 nm, with a proportion of micropore volume higher than 50% in all samples. The best hydrogen storage capacity at 1 bar and 77 K was equal to 1.57 wt% for the sample obtained from a 1:1 ratio of KOH/biochar. The same activated carbon showed an adsorption value of 5.5 wt% at 77 K and 80 bar, but it showed a total pore volume of 0.451 cm$^3$/g and a BET surface area of 1173 m$^2$/g.

### 2.1.3. ACs by Alternative Chemical Activation

Chemical activation of the carbon porous materials with potassium hydroxide is undoubtedly one the most used procedures in the field of chemically activated carbons. However, other compounds were also employed to chemically activate carbon materials. For example, Arshad et al. [112] produced AC from empty fruit bunch by combining KOH and CO$_2$ activations. The total surface areas (S$_{BET}$) of the produced ACs were in the range of 305–687 m$^2$/g, while the microporous structures increased, reaching a value up to 94%. Improvements of both the S$_{BET}$ and micropore volume (V$_{mic}$) were observed as the KOH loading increased. The hydrogen storage capacity was investigated at 77 K, under pressures from ambient to 100 bar. The hydrogen storage increased typically in the range of 0–20 bar and decreased for pressures > 20 bar. Empty fruit bunch activated with KOH (2.0 M) resulted in the best hydrogen uptake, with 2.14 wt% at 77 K and 19 bar.

Dogan et al. [164] prepared activated carbon samples by chemical and physical activations of tangerine peel, where KOH and ZnCl$_2$ were used as chemically activating agents. The carbon materials exhibited a porous structure, and their surface areas increased from being treated with the various concentrations of ZnCl$_2$ and KOH. The hydrogen-uptake values of the activated carbons produced by the combined effect of ZnCl$_2$ and KOH were higher than those obtained using KOH, probably due to the formation of a different porous substructure.

M. Sevilla et al. [155] reported the production of activated carbons from different precursors: glucose starch, cellulose, eucalyptus sawdust, and furfural. The first step of carbonization was performed by a hydrothermal process, and the cellulose-derived hydrochar was activated at temperatures in the range of 873–1073 K at hydrochar/KOH weight ratios of 1:2 or 1:4. The rest of the hydrochar materials were activated at 973 K and with hydrochar/KOH ratio of 1:4. Two of the cellulose-derived activated carbons were subjected to a double activation, i.e., repeating twice the same activation procedure, in order to further increase the surface specific area. Activated carbon materials with a high surface area (up to 2700 m$^2$/g) and narrow (0.7–2 nm, within the supermicropore range) size distribution were produced. The hydrogen adsorption capacities of the activated carbon materials oscillated from 2.1 to 2.5 wt% at 298 K and 1 bar; diversely, at 20 bar the uptake values fluctuated from 4.2 to 5.6 wt%. As for the samples prepared with a double activation step, adsorption values of up to 6.4 wt% were reached at 77 K and 20 bar.

Hu et al. [110] produced porous carbon materials for hydrogen storage, proposing a simple and effective method to transform sustainable biomass into porous carbon by means of partial lignin and hemicellulose degradation with an NaOH and Na$_2$SO$_3$ aqueous mixture. This strategy collapses the biomass structure to provide more active sites, avoiding the generation and accumulation of non-porous carbon nanosheets. The as-prepared sample showed both a high specific surface area (2849 m$^2$/g) and large pore volume (1.08 cm$^3$/g) almost completely on a micropore size. As a result, the as-prepared sample exhibited an appealing hydrogen storage capacity of 3.01 wt% at 77 K and 1 bar, or 0.85 wt% at

298 K and 50 bar, respectively. The value of hydrogen adsorption enthalpy is as high as 8.0 kJ mol$^{-1}$, which is superior to the most biochars. This strategy is of great significance concerning the conversion of biomass and the preparation of high-performance hydrogen storage materials.

Ganesan et al. [113] produced ACs from rice husk by means of sulfuric acid hydrolysis followed by low-temperature chemical activation with $H_3PO_4$. The final step of production involved heating at three different temperatures, 773, 973, and 1173 K, respectively, in a tubular furnace for 1 h, under continuous flow of argon gas. The best synthesized sample showed a value for hydrogen adsorption of 1.8 wt%, at 77 K and up to 10 bar.

Blankenship et al. [115] explored the potential of oxygen-rich activated carbons to increase the hydrogen storage capacity for the porous carbon materials. Their cellulose-acetate-derived carbons combine a high surface area and pore volume, i.e., 3800 m$^2$/g and 1.8 cm$^3$/g, respectively, mostly arising (>90%) from micropores, with an oxygen-rich content. The carbon materials exhibited an enhanced hydrogen uptake of 8.1 wt% at 77 K and 20 bar; this value rises to a total uptake of 8.9 wt% at 30 bar, with an exceptional volumetric uptake of 44 g/L and 48 g/L at 20 bar and 30 bar, respectively. The hydrogen storage capacity at 298 K reached values up to 1.2 wt% at 30 bar, and their isosteric heat of hydrogen adsorption was above 10 kJ mol$^{-1}$.

### 2.1.4. Doped or Modified ACs

As stated, ACs well fulfil many requirements to be good candidates for $H_2$ storage (high specific surface area, microporous structure with optimal pore size, and chemical and thermal stability), but hydrogen uptake can be increased by doping them with metals and/or modifying their surface. In the first case, we mean the inclusion within the carbonaceous matrix of metal particles (pre- or post-activation), while, in the second case, we mean the modification of surface functional groups to evaluate any changes in the porous structure or possible chemical interactions with hydrogen, thus improving the storage capacity of the support.

As early as 2010, Huang et al. carried out studies on AC chemically activated from lychee trunk, subsequently modified with the inclusion of palladium (Pd) particles and oxidation of the surface [114]. The high $S_{BET}$ achieved (3400 m$^2$/g) allowed storage of hydrogen up to 2.9 wt% at 77 K and 1 bar, while the effect of the presence of Pd was highlighted through measurements at 303 K and pressures up to 60 bar, showing, however, only a slight increase compared to the starting AC (0.5 vs. 0.4 wt%) even using a consistent quantity of metal (10 wt% Pd-AC). There are no details indicating if the increase in the amount of the adsorbed hydrogen due to metal loading corresponds to a chemisorption of $H_2$ and the consequent formation of the metal hydride. What was more interesting was the indication that emerged from the influence of the oxygen functional groups on the storage capacity which inhibit hydrogen adsorption due to the steric hindrance effect for the presence of acidic groups.

In 2015, a study by Rossetti et al. [165] summarized the effects of doping with different metallic elements (Pt, Pd, Rh, Ni, and Cu) carried out on a reference AC, deposited by conventional impregnation (using aqueous solutions of salts) and by Chemical Vapor Deposition (CVD) from various precursors. The $H_2$ storage capacity, investigated at 273 and 77 K in a pressure range of 1–100 bar, has been improved in the samples doped with Cu and Ni; in these cases, significant results have been observed (at 273 K), with a more than three-fold increase of the stored $H_2$ amount at 100 bar than with the AC without metal particles. Moreover, important indications have been given regarding the preparation and procedure method to obtain a good dispersion and optimize the spillover phenomenon. These results have represented a significant step forward and a reference for the experimental works in the next years.

Further experimental evidence of the aforementioned results has also emerged in other studies. The preparation and hydrogen adsorption testing of the ACs doped with transition metals (Cu, Ni, and Pd) were also investigated more recently by Ya'aini et al. [166],

carrying out measurements at different pressure conditions (3, 7, and 10 bar) and room temperature. The metal-doped AC samples showed better results compared to pristine and acid-treated ACs due to the catalytic action by the metal transition being able to facilitate the spillover mechanism, thus increasing the hydrogen uptake. The best result was shown using palladium (0.62 wt% at 10 bar), demonstrating high efficacy in the dissociative chemisorption of dihydrogen with a strong catalyst effect. Moreover, the presence of oxygen functional groups in the acid-treated ACs confirms the blocks of the pores resulting in a reduction in hydrogen storage capacity.

Recent attempts to improve hydrogen storage capacity were made by trying to make changes in the structure or surface of the materials, as in the case of the experiment conducted by Conte et al. [167], with the inclusion of copper metal particles within the matrix of ACs produced by amorphous cellulose. The starting material, in the form of flakes, was impregnated by adding a solution of copper gluconate, and the compound, after drying in vacuum, was subjected to activation in a $CO_2$ atmosphere. The copper-doped ACs developed showed a positive catalytic effect of the metal particles, during the formation of the porosity, up to 1% of metal concentration. The improvements in textural properties have influenced the hydrogen adsorption obtaining (at 77 K and 75 bar) values more than the theoretical value predicted by the Chahine Rule [161], with the best $H_2$ storage capacity being 3.91 wt%.

Finally, Zhang et al. showed in a recent publication a different and innovative method of preparing ACs from biomass, i.e., through the use of the vapors produced by pyrolysis as a source of carbon deposited with a calcium citrate template [168]. The developed porous carbon material has a high specific surface area of 1703 $m^2/g$, with hydrogen adsorption performances of about 1.5 wt% at liquid-nitrogen temperature and at atmospheric pressure. The research work highlighted an interesting novel method for the preparation of ACs from biomass, within a perspective of green and circular economy of processes.

Among the possible doping methods, the boronation processes need to be mentioned. While there are many examples of boron-doped carbon-based materials, such as nanotubes, graphene, graphene oxides, etc. [169–171], or boron-substituted carbon materials, synthesized by the carbonization of phenolic polymers (or others) as carbon sources [172–174], there are fewer attempts to apply this method on ACs produced from biomass, particularly with the aim of verifying their influence on the adsorption of hydrogen. In the literature there, is a study from 2012 on the development of high-surface-area boron-doped carbon materials produced via the CVD of a boron-doped carbon (BCX) layer onto CM-Tec-activated carbon [175].

The principal aim that is common to all works of boron doping is to increase the binding energy with hydrogen because, on pristine ACs, it is low, i.e., ~5 kJ/mol [176], and insufficient for achieving significant storage capacity above cryogenic temperatures. In fact, to enhance the gas–solid interaction, the binding energy needs to be increased to the 10–15 kJ/mol range [176]. For physisorption ends, it is possible to apply two main approaches: doping the carbon matrix with suitable elements [175] or tuning/developing a very narrow pore structure to facilitate hydrogen interactions between the two sides of the walls [177]. An interesting and innovative approach for future materials could be to combine both strategies to amplify the binding energy effect and overcome the current limits in hydrogen storage performances. An interesting reference hitherward is represented by a recent work of 2019 by Kopac and Kırca [82] in which, even if using bitumen coal (then, not biomass), the effects of ammonia and boron modification on the surface and hydrogen sorption characteristics of prepared ACs were investigated. The borax decahydrate ($Na_2B_4O_7 \cdot 10H_2O$) treatments of the ammonia-modified samples (in the range 0.025–0.075 M) affected the hydrogen uptake capacity in a positive way, with an increase up to 4.1 wt% at 77 K and 1 bar, due to the improvement in surface features, including the microporous structure and the interaction of nitrogen and boron with hydrogen.

## 3. Boronation Methods

Boron (B) is one of the most employed dopants for carbonaceous materials for several applications since its atomic radius is very similar to that of carbon. For this reason, the probability of boron atoms being introduced inside a carbon matrix is quite high. When inserted, it modifies its properties, accordingly to the electronic states of the carbonaceous surface [178]. A theoretical work from Kim et al. [179] stated that $H_2$ binding to a B site results in a stretching of the H-H bond and a partial charge transfer from the $H_2$ $\sigma$-bond to the empty localized $p_z$ orbital of boron atoms. The calculated value of the B-H binding energy is in the ideal range for adsorption and desorption of $H_2$ at room temperature. Due to these important pieces of evidence, the boronation of activated carbon has aroused an increasing interest for the production of new materials with enhanced hydrogen storage capabilities. Many different methods for the boronation of activated carbons are already present in the literature (Figure 6). The most classical ones involve quite extreme heat treatments under ultra-high vacuum conditions, with the use of different boronating agents to obtain B substitution. More recently, new methods have been implemented in order to achieve milder reaction conditions.

The reason for choosing a lower T for the synthesis resides in the fact that boron atoms start to be more volatile for T > 773 K and might desorb from functionalized carbon materials. On the other hand, too-low temperatures might hinder the boron substitution process. Independently from the selected synthetic methodology, B functionalization may be performed at different stages of the overall synthetic procedure. In the case of activated carbons, it can be performed during one of the following phases: (i) prior to the carbonization process; (ii) before the activation procedure, and (iii) post-activation, depending on the carbon precursor and on the selected functionalization procedure. In the case of synthetic carbon materials (nanotubes and graphene derivates), the B precursor can be delivered simultaneously to the C precursor in the desired ratio. Boron introduction inside carbonaceous materials' structures is fundamental not only for gas sorption applications but also for electrochemistry (electrode materials) and catalytic applications.

With this in mind, the most relevant boronation procedures from the literature are summarized in deeper detail in the following section [169].

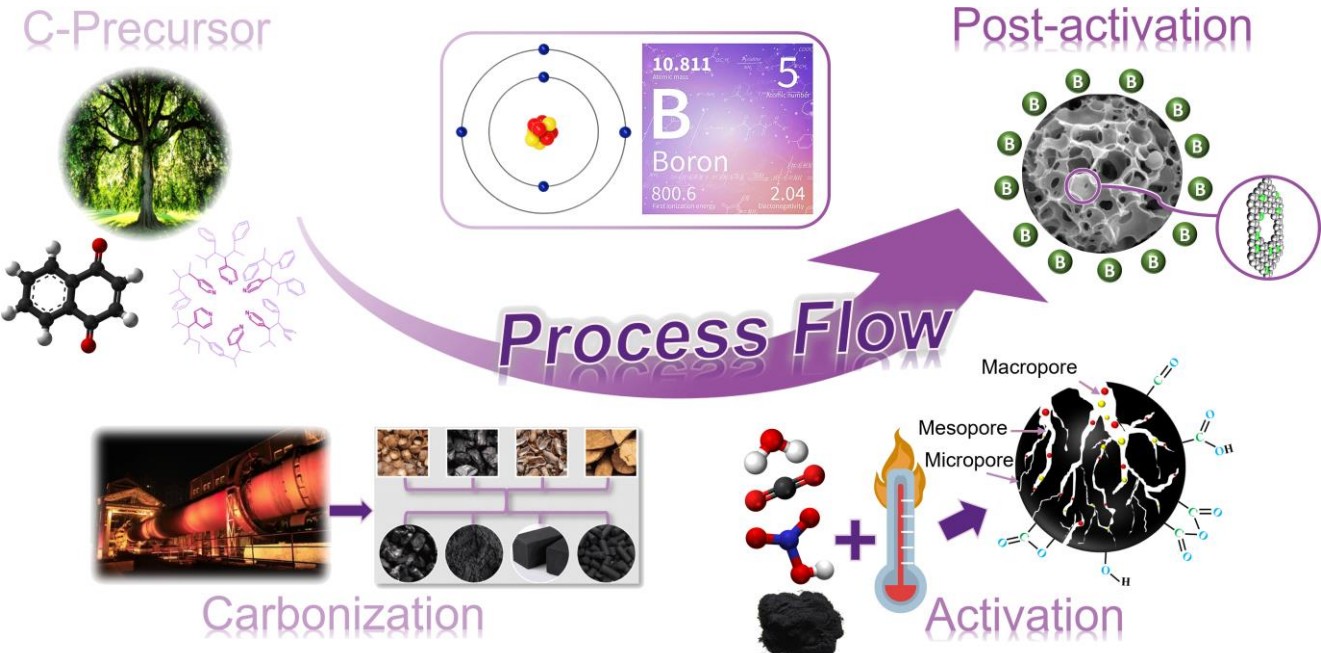

**Figure 6.** Sketch of different boronation techniques of ACs.

### 3.1. Classical Methods

Carbon Nanotubes (CNTs) are among the most versatile carbon-based materials, and the Chemical Vapor Deposition (CVD) method is one of the most used methods for their synthesis on a large scale [180,181]. In this technique, the precursors are introduced into a heated chamber, which is then brought to the desired temperature [182]. In the case of B-CNT, a proper boron precursor either in the solid [183], liquid [184], or gaseous [185] phase, is introduced together with the catalyst and the carbon source inside the furnace. Among the most used B precursors we found boric acid, triisopropyl borate, triethyl borate, trimethyl borate, and $B_2O_3$. Chen et al. [186] used Microwave Plasma–Chemical Vapor Deposition (MP-CVD) as synthetic method; the selected carbon source was $CH_4$, while, as the boron precursor, $B(OCH_3)_3$ was employed at various concentrations. Such a method was able to produce bundles of multi-walled CNTs, either pure or B-doped. This study showed that an increase in B concentration during synthetic procedure reduces the graphitization of B-CNTs, evaluated with the increase of $I_D/I_G$ ratio in Raman spectra of boronated samples, with respect to the undoped ones. In other words, the boron-doping effect strongly influences the crystallinity of CNTs. Despite the fact that the presence of boron was demonstrated by means of the *Secondary Ion Mass Spectrometry* (SIMS) technique, the authors did not report any quantification data. Mondal et al. [187] used a fixed bed reactor for CVD process, while $BF_3$/MeOH solutions (1.30 M, 1.04 M, or 0.58 M) were used as the source of B, and the reactor bed was packed with $Fe/Ca(BO_3)_2/CaCO_3$ which served either as catalyst or as an additional B source. In contrast to what was observed in the study reported above, the authors claimed that, in Raman spectra, the $I_D/I_G$ ratio decreased with increasing boron concentration, as confirmation that B atoms were introduced into the carbon layers, thus creating defects. In this work, the B amount measured by means of Maldi TOF-MS and laser ablation ICP-OES was <0.5 wt%, meaning that the combination of C- and B-precursors was not ideal for an effective boronation. Wang et al. [185] used the Electron Cyclotron Resonance–Chemical Vapor Deposition (ECR-CVD) [188,189], creating a microwave plasma that is similar to the MP-CVD technique. With such a methodology, they were able to produce both pure and B-doped CNT bundles by simply adding gaseous $B_2H_6$ in different ratios with respect to the carbon precursor (1:4, 1:2, 1:1, 2:1, or 4:1). Comparing doped and undoped materials, when boron started to be detected inside the CNTs (B/C ratio in precursor gas 1:1, with a boron functionalization degree of 4.2 wt%), the structure of CNTs changed from straight and tubular to bamboo-like. Furthermore, the outer diameters of the CNTs are larger, the tube walls are thicker, and the surface is rougher. What is interesting to notice is that, despite the fact that the ECR-CVD and MP-CVD techniques are similar in terms of their working principle, the B loading remarkably increased with ECR-CVD, reaching up 38.9 wt% against much lower values in the case of MP-CVD.

Ayala et al. [184] described the use of Hot-Wall Chemical Vapor Deposition (HW-CVD), a method involving high vacuum conditions. The most interesting feature is the use of a unique precursor for either the C or B source, namely pure triisopropyl orthoborate ($C_9H_{21}BO_3$), which was delivered through a horizontal plane into the furnace, together with $H_2$ as a gaseous reducing agent. The reaction temperature varied between 1063 and 1163 K, and the used catalyst consisted of several iron compounds supported on magnesium oxide. The authors succeeded in synthesizing B-doped CNTs with a narrow size distribution between 0.9 and 1.5 nm. The advantages of using a solvent-free single precursor method are evident; however, the authors did not provide B quantification for the produced samples, despite the fact that boron was clearly detected with the XPS technique. Nevertheless, in a similar study, Ruiz-Soria et al. were able to synthesize B-doped single-walled CNTs by High-Vacuum Chemical Vapor Deposition (HV-CVD) [190]. Starting from the same C and B precursor ($C_9H_{21}BO_3$) and similar temperatures (1123 K), they only obtained a B functionalization degree of 0.5 wt% (measured with XPS). It follows that even though the B:C ratio in triisopropyl orthoborate is 1:9, most of the B atoms were not kept inside the carbonaceous materials structures as they formed. Ceragioli et al. [183] and Ishii et al. [191] used Hot-Filament Chemical Vapor Deposition (HF-CVD), instead. As

starting materials, they both choose alcoholic solutions of B precursors: in the first case, ethyl alcohol and $B_2O_3$ (the B:C concentration ratio in the feed was 5000 ppm), whilst, in the second one, methanol and $H_3BO_3$ were employed (with 1.0 wt% of B in the solution). While in the first study B was not quantified, in the latter one, a B substitution of 1 wt% was achieved, meaning that almost all the boron atoms introduced were successfully held inside the nanotube structure.

The last work we report concerning the alcoholic solution of B-precursors is that from Preston and co-workers [192]. In this study, they produced B-doped few-walled CNTs, starting from different alcohol mixtures as the C source (pure MeOH or MeOH:EtOH solutions in either 1:1 or 3:1 ratio) at 1023 or 1123 K, while 2 wt% of diborane gas ($B_2H_6$) was used as the B precursor. Briefly, few-walled CNTs are multi-walled nanotubes that still exhibit a high aspect ratio and low defect density by possessing a wall number between two and six, although conserving structural integrity of the inner tubes. The choice of the catalyst was different with respect to the traditional Fe-based ones: indeed, they used Co:Mo:MgO (1:0.5:100) hybrid material pre-loaded inside the CVD chamber. In all the obtained samples, they were able not only to quantify the B content, but also a concentration gradient was found. Indeed, by means of Electron Energy Loss Spectroscopy (EELS) coupled with S-TEM, the detected B concentration in the nanotubes' "bulk" was quite lower than the one revealed at the nanotubes' "edge" (2.8 wt% vs. 7.2 wt%).

Arc discharge was among the first techniques employed for CNTs' preparation, and, being tremendously effective, it still remains a widely used technique to produce undoped [193] and B-doped CNTs. It consists of the passage of an electron current through an anode and cathode, both made of high-purity graphite, under an inert atmosphere. Aiming toward the doping of carbon nanotubes, a modified electric arc discharge process can be used, in which a mixture of both B precursor and carbonaceous material powders are placed inside a hollow anode, always under an inert atmosphere. In the work of Stephan et al. [194], a mixture of amorphous elemental B and graphite powder (1:2 weight ratio) was placed inside a hollow graphite anode; then the electric arc discharge (25 V and 100 A) was applied between the anode and a pure graphite cathode, under a $N_2$ atmosphere (200 mbar). Larger graphite sheets and nanotubes were obtained, with a maximum amount of B around 2 wt% (detected with EELS). However, the B-doping was not uniform, and co-doping with N atoms was observed. Furthermore, boron nitride formation was also detected, meaning that the employed procedure led to mixed products, in terms of either type of structure or heteroatom doping. A very similar procedure was employed by Maultzsch et al. [195] and Babanejad et al. [196]: indeed, also in these cases, the authors synthesized B-doped CNTs by introducing elemental B powder and graphite powder inside a hollow anode. Different from the previous case, reactants were suspended in ethanol, and the gas of choice was argon in place of nitrogen. By switching to a noble gas, N-doping was completely avoided. Furthermore, the boron concentration was more uniform and increased up to 4 wt%.

In an alternative method used by Carroll and co-workers [197,198], B-doped CNTs were obtained by means of a $BC_4N$ rod directly employed as an anode, working with graphite as a cathode, under helium pressure (600 Torr). The synthesized products showed a content of B between 1 wt% and 5 wt%, having a N content less than 1 wt%. Nevertheless, the synthesized samples presented a non-uniform boron distribution. Indeed, by means of Scanning Tunneling Microscopy (STM) and Scanning Tunneling Spectroscopy (STS), it was possible to identify local areas of high boron concentration. Through Local Density of States (LDOS) analysis of microscopy data, the authors detected a new state between valence and conduction bands, close to the Fermi level, compatible with the presence of island of $BC_3$ phases inserted in the walls of the nanotubes. Interestingly, despite the presence of the equimolar amounts of N and B in the source, nitrogen was present exclusively in traces amount. The arc discharge syntheses illustrated so far did not envisage the use of catalysts for nanotubes production. Differently, in the work presented by Wang et al. [199], a CoNiB amorphous alloy, placed inside the hollow anode for the arc discharge setup, acted either as the catalyst or as a boron source. Inert atmosphere was ensured through the use of 550 Torr

of a 9:1 mixture of He:$N_2$, and an electron current of 100 A was provided for the synthesis. In this way, single-walled B- and N-doped CNTs were obtained, characterized by a higher N content with respect to B (2.2 wt% and 1.2 wt%, respectively).

The laser ablation is a process in which a high-power laser focuses on a target, at high temperatures, to vaporize or sublimate the material of interest, which will react in high vacuum or onto a suitable support. Among the possible materials obtainable with such methodologies, it is possible to find pure and doped carbon nanotubes, even if they are not so common. Gai and co-workers [200,201] used laser ablation to obtain B-doped single-walled CNTs. Inside a quartz tube heated to 1373 K, they placed different targets composed of carbon paste mixed with a Co:Ni catalyst (0.5:0.5 wt%) and elemental boron (with different nominal B concentrations in the range of 0.5–10 wt%). The chamber was maintained under argon atmosphere, at a pressure of 500 Torr, while keeping a gentle flow of the same gas throughout the process; then the target was hit by a neodymium-doped yttrium aluminum garnet (Nd:YAG) laser (1064 nm, 10 Hz) for the ablation process. The authors observed substantial differences in the nanotubes' wall thickness depending on the B concentration in the target, passing from single-walled (with up to 2.5 wt% of B) to multi-walled (above 2.5 wt% of B). When the target contained 10 wt% of B, the authors observed the formation of large amounts (up to 60 wt%) of boron carbide ($B_4C$), with traces of amorphous B and $B_2O_3$, thus compromising the CNTs' yield. Exploiting the previous technology, Ayala et al. [202] used a similar procedure for a study of the superconductivity of this type of materials. The laser ablation method was focused on the target containing 1.5 wt% of B. This peculiar boron concentration in the target was of particular interest since it provided B-doped single-walled CNTs with a very narrow diameter distribution, strongly limiting $B_4C$ formation, which was almost nullified. Nevertheless, the boron content (4.5 wt%) was evenly distributed along the sample.

All the preparation techniques so far described require extreme conditions (high temperature and ultra-high vacuum) to obtain the desired samples, making them of difficult scalability and limited application. The high regularity in the carbon structures obtained and the doping level render them extremely attractive for electrochemical and superconductivity purposes (mainly electrodes production), albeit restricted to niche applications. With these procedures, boron is always introduced during the formation of the carbonaceous materials, making them, in principle, really efficient. However, for gas storage applications, requiring extremely larger volumes of sorbent materials, the scalability of the boronation processes cannot be further neglected. In addition, the regularity of the materials necessary for such use is less strict with respect to electrochemical ones, in which conductivity is strongly affected by the structural order of the materials employed. Finally, the sample amount is not the sole important parameter, since the materials' specific surface area is a strongly discriminating characteristic. For these reasons, substitution preparation techniques allow the production of much larger quantities of samples, using more affordable and simple setups, united with cheaper and much more abundant carbon precursors, including organic wastes (agricultural, forestry, and industrial). Furthermore, dealing with either biomass precursors or pre-synthesized activated carbons, these methods seem to be the best choice for materials preparation for a less specialized and bulkier application, even though still with limited reported cases.

### 3.2. Substitution Techniques

In the substitution reactions, the activated carbons or their precursors are treated with boronating agents in solution or in the solid state, and then subjected to heat treatments in an inert atmosphere for carbonization and activation. Liu et al. [203] used straw from recycled biomass as a carbon source and $H_3BO_3$ as a boron precursor. A carbon precursor and boric acid with a 1:1 weight ratio were dispersed in ethanol solution, dried, and then carbonized at 1173 K for 2 h under nitrogen flow. Subsequently, the obtained carbon was chemically activated by mixing it with KOH in a 1:1 mass ratio, and then it was calcined at 873 K for 2 h in a $N_2$ atmosphere. The obtained material exhibited a high specific surface

area (1190 m$^2 \cdot$g$^{-1}$) and a boron concentration of 4.3 wt%. Unfortunately, such material was not employed for gas sorption, but as support for catalytic applications. Moreover, Zhu and co-workers [204] prepared a boronated activated carbon as support for an Zn-based catalyst in acetylene acetoxylation. A H$_3$BO$_3$ water solution (5 g/L) was employed as a B source; however, this time, the boronation procedure was carried out on a pre-formed activated carbon at different temperatures, ranging from 773 to 1173 K. Moreover, in this case, the surface area value was maintained extremely high (up to 1015 m$^2 \cdot$g$^{-1}$), but the B content decreased down to 2 wt%.

Yao et al. [205], instead, used a co-doping procedure of B and N, in which the precursors were, respectively, (NH$_4$)$_2$SO$_4$ and B$_2$O$_3$ (added separately or in a 10:1 weight ratio, respectively). The boronation procedure was performed by mixing N and/or B sources in the solid state with an already prepared porous carbon powder [206] and then carbonizing at 1173 K under an inert atmosphere. The porous properties of the final material were enhanced by the presence of nitrogen atoms: indeed, the specific surface area grew from 650 m$^2 \cdot$g$^{-1}$ in the case of B-doped carbon to 880 m$^2 \cdot$g$^{-1}$ for the co-doped sample. The synergic effect of the co-doping was evident on the boron content, as well, increasing from 1.0 wt% for the boron containing sample to a maximum amount of 6 wt% in the case of the B/N co-doped one [207].

The procedure of Wee et al. [208] involved a heat treatment directly onto the solid-state mixture of activated carbon with 5 wt% B$_2$O$_3$ powder at a temperature between 1673 and 1873 K for 30 min, using argon as inert gas. The presence of boron at such high temperatures favored B mobility with the consequent obstruction of the material pores' entrance, thus provoking a tremendous drop in the specific surface area of the material, passing from 1780 m$^2 \cdot$g$^{-1}$ for the undoped sample, to 955 m$^2 \cdot$g$^{-1}$ for the one activated at 1673 K, and finally to 27 m$^2 \cdot$g$^{-1}$ for the doped carbon activated at 1873 K. Such contraction happened without any particular drop in B content, decreasing from 1.8 at% to 1.4 at%. This paper evidenced that B content is not the only valid parameter to discriminate whether a material is suitable for hydrogen storage (and more generally specific gas sorption) application or not.

While the studies herein reported deal with boronated activated carbons derived from biomass wastes, both Li et al. [209] and Ariharan et al. [173] used chemicals (resorcinol and formaldehyde) as a carbon source. However, in the first procedure, the authors used a templating agent, as well, namely Pluronic F127, while no template was employed in the second process. Additionally, the boron precursors differed: indeed, in the first case, H$_3$BO$_3$ was employed (in B-to-C molar ratios of 0.025, 0.05, 0.075, and 1), while in the second one, triethyl borate was used (in B-to-C molar ratio of 0.3). In both syntheses, carbonization was performed after mixing C and B precursors, but neither physical nor chemical activation was performed. In his work, Li et al. obtained a maximum boron functionalization percentage of 1.3 wt%, but they kept the specific surface area value of the samples above 600 m$^2 \cdot$g$^{-1}$. Conversely, in the paper of Ariharan et al., despite the fact that the boron substitution value was around 12 wt%, the surface area of the sample strongly decreased down to 65 m$^2 \cdot$g$^{-1}$.

The synthetic protocols, either for classical or substitutional boronation methods, need temperatures of 873 K or above at a certain point of the procedure. The last substitution technique published by Y. Shin et al. [210] described an innovative procedure under mild experimental conditions. This process, called the "wet procedure", involved a reaction in THF, using a BH$_3 \cdot$THF adduct as the B precursor, which was left to react at 353 K in the presence of a high-surface-area graphene oxide for several days (3, 5, or 7), without the need of a final heat treatment. The best material was obtained after 7 days of reaction, displaying a B substitution up to 2 wt% and a surface area value of 180 m$^2 \cdot$g$^{-1}$.

The above-cited results concerning, to the best of our knowledge, all the reported carbon-based materials' boronation procedures, are summarized in Table 3. Concerning the most common experimental conditions, in the majority of the reported cases, H$_3$BO$_3$ was the most employed boron precursor for substitution processes, followed by B$_2$O$_3$, elemental B, and organo-boron compounds. The temperatures of the heat treatments ranged between

773 and 1873 K, with percentages of boron substitution going from 1% up to a maximum of 12.7% in the best case.

**Table 3.** Summary of the boronation procedures reported in this section.

| Entry | Carbon Type | Synthesis | Boron Doping | Boron Amount (wt%) | Refs. |
|---|---|---|---|---|---|
| 1 | | MP-CVD | B(OCH$_3$)$_3$; in situ | n.q. [†] | [186] |
| 2 | MWCNTs | CVD | BF$_3$-MeOH; in situ | <0.5 | [187] |
| 3 | | ECR-CVD | B$_2$H$_6$; in situ | 4.2–38.9 | [185] |
| 4 | | CVD | | 2.8 | [192] |
| 5 | | HW-CVD | | n.q. [†] | [184] |
| 6 | SWCNTs | | C$_9$H$_{21}$BO$_3$; in situ | n.q. [†] | [211] |
| 7 | | HV-CVD | | <0.5 | [190] |
| 8 | MWCNTs | HF-CVD | B$_2$O$_3$/EtOH; in situ | n.q. [†] | [183] |
| 9 | | | H$_3$BO$_3$/MeOH; in situ | 1.0 | [191] |
| 10 | | | | 2.0 | [194] |
| 11 | MWCNTs | Arc Discharge | Elemental B | 0.5–4.0 | [195] |
| 12 | | | | n.q. [†] | [196] |
| 13 | SWCNTs | Arc Discharge | BC$_4$N | 1.0–5.0 | [197,198] |
| 14 | | | CoNiB alloy | 0.97–1.2 | [199] |
| 15 | SWCNTs | Laser Ablation | Elemental B | 0.5–10.0 | [200,201] |
| 16 | | | | 4.5 | [202] |
| 17 | | | H$_3$BO$_3$; before Carbonization | 4.3 | [203] |
| 18 | | | | 1.3 | [209] |
| 19 | Activated Carbon | Solid-Phase Substitution | B$_2$O$_3$; post-activation | 1.0–6.0 | [205] |
| 20 | | | | 1.4–1.8 | [208] |
| 21 | | | H$_3$BO$_3$; post-activation | 2.0 | [204] |
| 22 | | | B(OEt)$_3$; before carbonization | 12.7 | [173] |
| 23 | Graphite oxide | Liquid-Phase Substitution | BH$_3$-THF; post-activation | 0.9–2.4 | [210] |

[†] Boron amount not quantified.

Such variability is due to the intrinsic nature of carbon sources: biomasses largely employed as activated-carbon precursors may require extremely different processes, as well as activation and post-activation procedures [212]. However, this is not necessarily a weak point, as it guarantees almost an infinite range of possible combinations for the boronation of carbonaceous porous materials.

Once the materials are synthesized/functionalized in the presence of boron, dedicated physicochemical characterization techniques must be employed in order to reveal the effectiveness of the preparation methodologies.

## 4. Characterization Techniques for Boron-Doped ACs: An Overview

Due to their intrinsic heterogeneity and complexity, the physicochemical characterization of carbon-based materials often represents a challenge. Nevertheless, throughout the years, several protocols have been implemented depending on the typology of material that needs to be described [213–216]. In effect, for more regular structures (such as graphene and graphene-related materials) [217–219], there will be a stronger focus on structural techniques, whereas for more irregular ones (e.g., nanotubes and activated carbons), the attention will move towards textural and morphological techniques [220,221]. The peculiar case of B-doped carbons requires techniques that are able not only to fully portray the overall characteristics (structural, textural, morphological, and superficial) of the activated carbons [222] but also to provide information about the presence and the status (i.e., the prevalent oxidation state) of boron species in carbonaceous materials (Figure 7) [169,223].

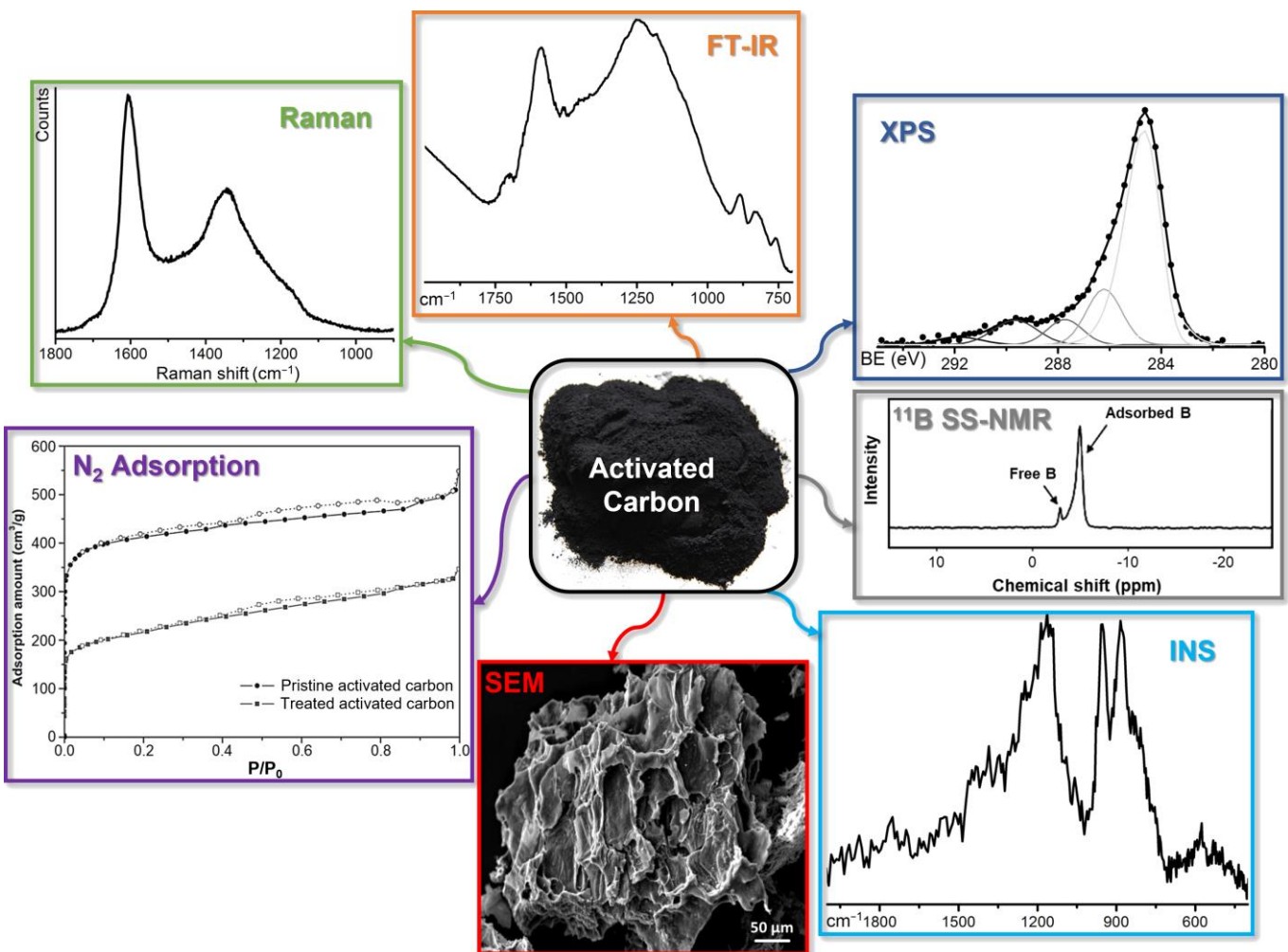

**Figure 7.** Overview of the principal characterization techniques mostly employed for boronated carbonaceous materials.

Boronation procedures, either in wet or in dry conditions, can provoke a severe modification on the textural properties of activated carbons, which can be detected with a direct observation of the materials by means of electron microscopies (especially SEM). However, microscopies in this case are not informative about boron insertion in the carbonaceous matrix, even with EDX elemental analysis, since B is indistinguishable from C with such a technique. In addition, the amount of boron usually loaded into carbonaceous materials is less than 5%, making it difficult to detect. Despite this, more meaningful evidence of boron insertion might come from more specific physicochemical characterization techniques.

Surface area analyses obtained by N₂ adsorption at 77 K, combined with *ad-hoc* mathematical models [224,225], are able to provide good evidence for eventual textural modifications of the materials, leading to an expansion or a contraction of their total surface area [226]. In the specific case of boron-doped carbons, a clustering phenomenon causing a progressive pore occlusion has been reported [227], thus reducing the total surface area and the pore volume, as well [208]. However, this proof alone is not sufficient to confirm B-doping.

Despite being usually poorly crystalline, almost the totality of activated carbons can be described as a stack of graphene platelets (containing different amounts and types of defects), therefore presenting a certain regular spatial distribution along at least one direction perpendicular to the platelets [214]. Thus, X-ray diffraction (XRD) can, in principle, monitor the interplanar distance between materials planes [228]. However, the extremely

low level of B-doping makes it relatively difficult to observe such phenomenon, despite interplanar boronation having been reported [229].

More solid proofs for boron insertion and (above all) its speciation can be gathered by Solid-State Nuclear Magnetic Resonance (SS-NMR). Indeed, [11]B SS-NMR can provide structural information on materials strongly missing long-range order, as in the case of activated carbons [230,231]. With this in mind, it is potentially possible to distinguish between species adsorbed onto the graphenic planes constituting the materials [229,232,233] and the ones that are chemically bound to them [234]. Along with the technique being quite sensitive to the local environment of B atoms, it should also be able to discriminate among different preferential functionalization spots on the carbon surface.

Even though materials' functionalization takes place principally on their surface, the use of techniques which are more surface-sensitive is mandatory. Nowadays, the most sensitive and reliable technique for chemical investigation of materials surface is X-ray Photoelectron Spectroscopy (XPS), detecting photoemitted electrons from materials first 10 nm below the surface. XPS provides access to the entire chemical composition of materials, identifying different elements through their electrons binding energy. In this way, by observing eventual energy shifts of the emitted electrons, it is also possible to observe eventual differences in the oxidation state and chemical bonding of such atoms. This kind of information is extremely powerful, allowing us not only to probe for the presence of certain elements but also to discover their chemical environment. Correlations among elements are visible through peak broadening [208,235] or through the appearance of multiple peaks in the specific B 1s spectral region [236,237]. The ratio between same elements differently bound is obtainable through peak deconvolution: in the case of boron, it is possible to identify whether it is connected to either O, C, or H depending on its binding energy [238–240].

Finally, it is possible to acquire further surface and structural information from other spectroscopic techniques, especially vibrational ones. Among them, Raman spectroscopy applied to carbon-based materials is the one that provides a deeper insight into the structural organization of the materials. The typical Raman spectrum of activated carbons displays two main bands: the first one around 1600 cm$^{-1}$ (G band), due to the stretching of sp$^2$ C–C bonds with E$_{2g}$ symmetry [241]; and the second one around 1350 cm$^{-1}$ (D band), assigned to a lattice breathing mode with A$_{1g}$ symmetry. Such a vibration is forbidden in graphite crystals but becomes Raman-active in the presence of structural disorder [242]. The ratio between the intensity of these two bands (Equation (2)),

$$R = I_D/I_G, \tag{2}$$

usually correlates with the degree of structural disorder in the material: the higher the ratio, the higher the structural disorder is. When dealing with boron doping, the R value tends to increase after boronation procedures, meaning that Raman spectroscopy can be used as an indicator of successful functionalization [208,243].

Infrared spectroscopy (FTIR) is also sensitive to chemical composition and bonding; however, the strong radiation absorbing character of carbonaceous materials forces the investigation towards diluted systems (usually in KBr) [215] in order to appreciate eventual modifications of the materials. Furthermore, such a technique is selective for those vibrational modes associated with a modification of the dipole moment between the atoms involved [244]. In this way, although with low intensity, it is possible to discriminate chemical species constituted by heteroatoms, meaning that the largest is the difference in electronegativity in respect to carbon, and the stronger is the IR signal. Following this rule, the C−B bond (centered on 1020 cm$^{-1}$ [245–248]) should be, in principle, more IR-active than C−H bonds and therefore fairly well visible in the case in which it is formed [249–252].

The last technique employed in the characterization of boronated carbonaceous materials is Inelastic Neutron Scattering (INS). This is an extremely powerful tool for detecting vibrations involving hydrogen-containing species populating the edges of carbon-based materials [253,254], since the incoherent scattering cross-section of H is 80.26 barn, against

0.001 barn of C [255]. Furthermore, unlike other vibrational techniques, INS have no selection rules. This means that the signal's intensity is uniquely proportional to the amount of each chemical species present under the neutron beam, making it an extremely potent quantitative technique. The main drawbacks of the INS are the need for a gram-scale amount of sample to be fit under the neutron beam and the need of large-scale facilities for neutron production, making this technique extremely costly and accessible only after proposals evaluation. Despite the enormous advantages described, boron has an ambiguous behavior when interacting with neutrons. Although its average incoherent scattering cross-section is 1.70 barn, B is also one of the most neutron-absorbing materials [256]. This characteristic may potentially interfere with the incoherent scattering phenomenon. However, when dealing with B-doping of only a few wt%, it is still possible to make a direct measurement of the vibrational properties of C-B bonds [257]. Alternatively, dealing with an element having a strong affinity for molecular hydrogen, it is still possible to indirectly prove its presence by dosing $H_2$ at cryogenic temperatures to observe B-H bonding [258], also with the possibility to evidence spillover species that can be stored on the carbon surface [259].

## 5. Summary and Perspectives

In this review, we explored the various possibilities and limitations provided by current hydrogen storage methodologies. We are in the presence of a strong unbalance between compressed and cryo-based hydrogen storage technologies and physical/chemical sorption methods. The first ones, on the one hand, allow the stable warehousing of huge quantities of $H_2$ in an economically convenient way, while on the other hand, in the case of storage failure, the risk of catastrophic events makes such technologies extremely dangerous, especially when applied to mobility applications. Sorption methods are much safer, thus making them preferable when dealing with transportation. Nevertheless, there are several drawbacks that still need to be overcome. Firstly, the storage capacity of these methods (especially the physical ones) is far lower with respect to high-pressure storage. Secondly, the kinetics of $H_2$ release needs to be quick and, especially in the case of chemically stored hydrogen, this might be a limiting factor. While for the kinetic issue of chemisorbed hydrogen, there is close to zero possibility of improvement, in the case of physisorbed $H_2$, there is room for improvement.

In particular, herein we explored carbon-based materials (preferably coming from organic waste recovery) as hydrogen storage systems due to their lightness and large surface area. However, carbon does not have preferential interactions towards hydrogen, meaning that the $H_2$ adsorption capacity on carbon-based materials depends on several factors, such as surface porosity, surface functionality, structural properties, and heteroatoms content [146]. Indeed, the crystal structure of carbonaceous materials is a pivotal parameter, since carbon's crystallinity enhances the molecular hydrogen physisorption capacity, trapping $H_2$ molecules in between graphitic platelets.

This review, instead, was focused on the influence that doping elements have on storage properties, narrowing our choice on boron. Its atomic dimensions allow it to substitute carbon atoms almost to any extent in materials (we documented B-loading up to 40%) [185], and its preferential affinity towards hydrogen molecules should tremendously affect the storage capacity of the sorbent materials. The drawback of boronation lies in the quantity of obtainable doped material: high B content is achieved almost only for CNTs, that are produced on the grams scale; when dealing with activated carbons (which are produced in the hundreds of tons scale), the B doping usually does not exceed 4–5% [205], thus strongly limiting its influence on the storage capacity.

Although the boron loading is low for activated carbons, the parameters that can be varied along the synthetic route are so many (both in the starting carbon materials and in the B precursors) that, optimistically, in the close future, an effective, scalable, and affordable procedure might be found.

**Author Contributions:** Conceptualization, M.C., A.P. and G.C.; writing—original draft preparation, A.L., A.M., R.C. and O.D.L.; writing—review and editing, A.L., A.M. and A.A.; supervision, M.C. and A.P. All authors have read and agreed to the published version of the manuscript.

**Funding:** This research received no external funding.

**Data Availability Statement:** Data sharing is not applicable to this article.

**Conflicts of Interest:** The authors declare no conflict of interest.

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
