# Peer review of "Boronation of Biomass-Derived Materials for Hydrogen Storage"

_compounds, doi:10.3390/compounds3010020_

Round 1

Reviewer 1 Report

This paper focused on the boronation of biomass-derived materials around the application of hydrogen storage. Firstly, they introduced the benefits and drawbacks of chemisorption or physisorption for H2 storage, then, systematically described the physical and chemical factors for governing the hydrogen absorption such as specific surface area, pore size, volume and N,O doping etc. Secondly, the physical activation (CO2 and/or steam) and chemical activation (ZnCl2, KOH, NaOH etc.) were emphasized that help improving carbons (biomass derived) textural properties for hydrogen storage. Finally, this paper gave an insight into boronation methods which narrowing on boron doping for carbonaceous materials modified. This review is logic clear, sentences smooth, and with concise figures, had a certain effect on the development of the field of hydrogen storage by using biomass materials. I suggested it to published after some minor revisions.

(1)    A one-to-one correspondence between references and descriptions in table 1 should be better, please refer to table 2 and table 3 in this article.

(2) The all figures have no source be quoted, it should be quoting the original sources, books and journal articles on the topic, even if the picture is rearranged or recreated by yourself.

(3)   Some important references in hydrogen storage fields should be cited:

ACS Materials Lett. 2022, 4, 967-977; Cell Rep. Phys. Sci. 2021, 2, 100289; Nano Res. 2020, 13, 105.

Author Response

Comments and Suggestions for Authors

This paper focused on the boronation of biomass-derived materials around the application of hydrogen storage. Firstly, they introduced the benefits and drawbacks of chemisorption or physisorption for H2 storage, then, systematically described the physical and chemical factors for governing the hydrogen absorption such as specific surface area, pore size, volume and N,O doping etc. Secondly, the physical activation (CO2 and/or steam) and chemical activation (ZnCl2, KOH, NaOH etc.) were emphasized that help improving carbons (biomass derived) textural properties for hydrogen storage. Finally, this paper gave an insight into boronation methods which narrowing on boron doping for carbonaceous materials modified. This review is logic clear, sentences smooth, and with concise figures, had a certain effect on the development of the field of hydrogen storage by using biomass materials. I suggested it to published after some minor revisions.

(1)    A one-to-one correspondence between references and descriptions in table 1 should be better, please refer to table 2 and table 3 in this article.

The references cited in the caption of Table 1 were added inside the Table itself, accordingly to Tables 2 and 3.

(2) The all figures have no source be quoted, it should be quoting the original sources, books and journal articles on the topic, even if the picture is rearranged or recreated by yourself.

All the Figures present in this paper were originally designed and produced by the authors of the paper. Therefore, no citation was added in the caption of the Figures.

(3)   Some important references in hydrogen storage fields should be cited:

ACS Materials Lett. 2022, 4, 967-977; Cell Rep. Phys. Sci. 2021, 2, 100289; Nano Res. 2020, 13, 105.

The suggested references were included in the section regarding hydrogen storage with numbers (48-50).

Reviewer 2 Report

1. In Figure 2, there are three physical hydrogen storage methods. However, in section 1.2, LH2 and CcH2 are not introduced respectively. In particular, CcH2 is currently regarded as a very excellent hydrogen storage method. Please refer to Zhang J, Yan Y, Zhang C, et al. Properties improvement of composite layer of cryo-compressed hydrogen storage vessel by polyethylene glycol modified epoxy resin[J]. International Journal of Hydrogen Energy, 2023, 48(14): 5576-5594.     Zhao X, Yan Y, Zhang J, et al. Analysis of multilayered carbon fiber winding of cryo-compressed hydrogen storage vessel[J]. International Journal of Hydrogen Energy, 2022, 47(20): 10934-10946.

2. The author should standardize the citation format. If other people's images are cited in your figures, please provide references. For example, Figure 6 and Figure 7.

Author Response

Comments and Suggestions for Authors

  1. In Figure 2, there are three physical hydrogen storage methods. However, in section 1.2, LH2 and CcH2 are not introduced respectively. In particular, CcH2 is currently regarded as a very excellent hydrogen storage method. Please refer to Zhang J, Yan Y, Zhang C, et al. Properties improvement of composite layer of cryo-compressed hydrogen storage vessel by polyethylene glycol modified epoxy resin[J]. International Journal of Hydrogen Energy, 2023, 48(14): 5576-5594. Zhao X, Yan Y, Zhang J, et al. Analysis of multilayered carbon fiber winding of cryo-compressed hydrogen storage vessel[J]. International Journal of Hydrogen Energy, 2022, 47(20): 10934-10946.

The authors added a section (Section 1.3) in the text of the paper describing such storage methodology. Here is reported the added text:

1.3. Cryo-compressed gas

Cryo-compressed hydrogen storage has been introduced to overcome the disadvantages of both traditional storage methods mentioned above (Sections 1.1 and 1.2) by combining their main characteristics [27,28]. It occurs at cryogenic temperatures (20-50 K) on a pressurized vessel, although not as much as with the compressed one (≈ 35 MPa) [29].

Compared to traditional methods, cryo-compressed H2 storage presents some ad-vantages such as higher energy density [30,31], gravimetric capacities and volumetric efficiency [32], fast filling speed and high-pressure resistance [29,33], a reduced boil-off effect [34], and consequently reduced in-vessel over-pressurization and longer thermal endurance [35]. Additionally, it is a versatile method since cryo-compressed storage tanks are designed to endure both very low temperatures and very high pressures [36].

Nevertheless, there are still some limitations preventing this technology from be-coming commercially viable. In particular, cryo-systems are usually complex and hard to implement, requiring the careful and permanent management and monitoring of their thermal insulation levels [32] as they needs high energy for operation, they have considerable maintenance costs and a short no-loss unused period [37].

  1. The author should standardize the citation format. If other people's images are cited in your figures, please provide references. For example, Figure 6 and Figure 7.

All the Figures present in this paper were originally designed and produced by the authors of the paper. Therefore, no citation was added in the caption of the Figures.

Concerning the references, the authors employed the citation format for Articles, Books, and Book Chapters, directly from the journal website to be uniformed with MDPI editorial guidelines.

Reviewer 3 Report

The manuscript entitled: "Boronation of biomass-derived materials for hydrogen storage" written by A. Lazzarini, A. Marino, R. Colaiezzi, O. De Luca, G. Conte, A. Policicchio, A. Aloise, and M. Crucianelli presents an interesting review, which in my opinion, will be attractive to a wide range of readers. I recommend publishing it after minor revision:

1. Molecular hydrogen can also be stored in underground reservoirs. Please add this information. 

2. Lines 64-66: the sentence " To address the current..." is hard to understand. Please rewrite it.

3. In some places, like in line 150, Fig. 4, the capitol letters are not required. Please change it.

4. Line 385: "were" should be written instead of "where".

5. Line 885: abbreviation should be omitted.

6. Please rewrite the sentence in lines 909-911 ("To summarize, ...").

7. It would be good to add information about the position of C-B bands in the FTIR spectrum.

Author Response

The manuscript entitled: "Boronation of biomass-derived materials for hydrogen storage" written by A. Lazzarini, A. Marino, R. Colaiezzi, O. De Luca, G. Conte, A. Policicchio, A. Aloise, and M. Crucianelli presents an interesting review, which in my opinion, will be attractive to a wide range of readers. I recommend publishing it after minor revision:

  1. Molecular hydrogen can also be stored in underground reservoirs. Please add this information.

The authors added a sentence in the paragraph 1 (lines 69-74) of the paper describing such storage methodology. Here is reported the added text:

A first method, where there are suitable and available sites, a mid/long term storage of larger volumes of hydrogen it's possible using geological formations, such as subsurface depleted oil and gas reservoirs, aquifers, or cavern storage [12,13]. Instead, for more immediate application and fast release processes, the most common options are its compression as a gas, its condensation as a liquid, storage as cryo-compressed gas…

  1. Lines 64-66: the sentence " To address the current..." is hard to understand. Please rewrite it.

We rephrased the sentence as follows: “Having a solid and reliable storage technologies for different applications is a pivotal factor to satisfy the current and also the future demands of the hydrogen energy market [1,2]”

  1. In some places, like in line 150, Fig. 4, the capitol letters are not required. Please change it.

We checked for unwanted capital letters, and corrected when necessary.

  1. Line 385: "were" should be written instead of "where".

We corrected this typo.

  1. Line 885: abbreviation should be omitted.

We corrected this typo.

  1. Please rewrite the sentence in lines 909-911 ("To summarize, ...").

We rephrased the sentence as follows: “Concerning the most common experimental conditions, in the majority of the reported cases H3BO3 was the most employed boron precursor for substitution processes, followed by B2O3, elemental B, and organo-boron compounds.”

  1. It would be good to add information about the position of C-B bands in the FTIR spectrum.

The authors added the required information as follows: “Following this rule, C-B bond (centered around 1020 cm-1 [245-248]) should be in principle more IR-active than C-H bonds, therefore fairly well visible in the case it is formed [249-252].”